



# Comparison of observed borehole temperatures in Antarctica with simulations using a forward model driven by climate model outputs covering the past millennium

Zhiqiang Lyu[1], Anais J. Orsi[2], Hugues Goosse[1]

[1]Université catholique de Louvain (UCLouvain), Earth and Life Institute (ELI), Georges Lemaître Centre for Earth and Climate Research (TECLIM), Place Louis Pasteur, B-1348 Louvain-la-Neuve, Belgium
[2.]Laboratoire des Sciences du Climat et de l'Environnement (IPSL/CEA-CNRS-UVSQ UMR 8212), CEA Saclay, 91191 Gif-sur-Yvette CEDEX, France

**Correspondence:** *Zhiqiang Lyu (zhiqiang.lyu@student.uclouvain.be)*

**Abstract.** The reconstructed surface temperature series from boreholes in Antarctica have significantly contributed to our understanding of centennial and multi-decadal temperature changes and thus provides us a good way to evaluate the climate model ability to reproduce low-frequency climate variability. However, up to now, there were no systematic model-data comparisons based on temperature from boreholes at regional or local scale in Antarctica. Here, we discuss two different ways to perform such a comparison using boreholes measurements and the corresponding reconstructions of surface temperature at West Antarctic Ice Sheet (WAIS), Larissa, Mill Island and Styx in Antarctica. The standard approach is to compare climate model outputs at the grid point closest to each site with the reconstructions in the time domain derived from the direct borehole temperature observations. Although some characteristics of the reconstructions, for instance the non-uniform smoothing, limit to some extent the model-data comparison, several robust features can be evaluated. In addition, a more direct model-data comparison based on the temperature measured in the boreholes is conducted using a forward model that simulates explicitly the subsurface temperature profiles when driven with climate model outputs. This comparison in the depth domain provides many consistent signals with those in the time domain, but also suggest some information that we cannot extract from the comparison in the time domain. The major results from these comparisons are used to define some metrics derived from the borehole temperature data for future model-data comparison, and demonstrate the spatial representativity of the sites chosen for the metrics. The long term cooling trend in West Antarctica from 1000 to 1600 CE (-1.0 °C) is generally reproduced by the models, but often with a weaker amplitude. The 19th century cooling in the Antarctic Peninsula (-0.94°C) is not reproduced by any of the models, which tend to show warming instead. The trend over the last 50 years is generally well reproduced in West Antarctica and at Larissa (Antarctic Peninsula), but overestimated at other sites. The wide range of simulated trends indicates the importance of internal variability on the observed trends, and show the value of model-data comparison to investigate the response to forcings.



## 1.   Introduction

Although most of the world has been steadily warming over the last few decades, the temperature trend in Antarctica is
not homogeneous (Jones et al., 2016). Several syntheses relying on instrumental air temperatures records have shown
a large recent warming over the AP and parts of West Antarctica, but the trend for the other parts of the Antarctic
continent remains less clear (Chapman and Walsh, 2007; Nicolas and Bromwich, 2014; Steig et al., 2009; Turner et
al., 2005). The sparse instrumental data and the series covering generally less than 60 years do not allow to
characterize well the large interannual to multi-decadal variability at high southern latitudes. The mechanisms at the
origin of recent changes are thus still uncertain (Goosse et al., 2012; Jones et al., 2016; Nicolas and Bromwich, 2014).
Proxy-based reconstructions offer the opportunity to place the recent temperature changes in a longer context. Thanks
to their relatively good spatial coverage and their high resolution, the reconstructions based on water stable isotopes
derived from ice core have provided important information on temperature variability during past two millennia over
Antarctica. They indicate a significant cooling trend during the preindustrial period across all Antarctic regions and
confirm the strong spatial heterogeneity of the recent warming (Goosse, 2012; Schneider et al., 2006; Stenni et al.,
2017). However, the link between the isotope records and local climate is complicated, and this introduces significant
uncertainties in the reconstructions (Stenni et al. 2017, Klein et al., 2019).
Borehole temperature observations provide another opportunity to reconstruct surface temperature and several
studies have demonstrated their interest, particularly over Antarctica (i.e. Barrett et al., 2009; Muto et al., 2011; Orsi et
al., 2012; Zagorodnov et al., 2012; Roberts et al., 2013; Yang et al., 2018). Since the variable measured in the borehole
is the temperature itself, i.e. the variable that is reconstructed, no calibration is required against independent
climatologic data such as instrumental data. Nevertheless, the characteristics of heat conduction that blurs the surface
temperature history makes the reconstruction mathematically undetermined. Several approaches have been proposed
to overcome the problem as synthesized in Orsi et al (2012), for instance the Bayesian Reversible Jump Markov chain
Monte Carlo (Dahl-Jensen et al., 1999), the generalized least-squares inversion (Muto et al., 2011; Orsi et al., 2012;
Yang et al., 2018), and the Tikhonov regularization method (Roberts et al., 2013). By applying these methods, the
reconstructed temperature series have presented evidence of the existence of cold conditions corresponding to a Little
Ice Age in West Antarctica from 1300 to 1800 CE (Orsi et al., 2012), as well as of a recent warming trend in West
Antarctica (Barrett et al., 2009; Orsi et al., 2012; Yang et al., 2018), at some high elevation sites of the East Antarctica
(Muto et al., 2011; Roberts et al., 2013) and over the Antarctica Peninsula (Zagorodnov et al., 2012), though the timing
and magnitude vary between regions.
The reconstructed temperatures based on isotopic composition have been compared to results of climate models.
Most models display a relatively large and homogenous warming over Antarctica since 1850, which is inconsistent
with the signal inferred from the isotope records (Goosse et al., 2012; Klein et al., 2019; Stenni et al., 2017, Abram et
al. 2016). This disagreement may be due to the uncertainties in the reconstructions, or due to the uncertainties in the
climate models that may overestimate the response to greenhouse gas forcing or underestimate the natural climate
variability in the region (Jones et al., 2016; Neukom et al., 2018). However, a recent study assessing the link between



isotope record from ice cores and regional climate over Antarctica using pseudoproxy and data assimilation
experiments has not been able to identify any systematic bias in reconstructions on continental scale temperatures
based on $\delta^{18}O$ (Klein et al., 2019).

Up to now, there were no systematic model-data comparison for temperature reconstructed from boreholes at

regional or local scale in Antarctica. This is, on the one hand, due to the characteristics of the inversion that imposes
smoothing on a time window that increases as we go back in time and makes the comparison with the simulated
surface temperature difficult (Beltrami et al., 2006; Harris and Gosnold, 1999). Additionally, some reconstructions
have an uncertainty range of the same magnitude as the full variability provided by the climate model results, which
seriously limits the interest of data-model comparison.

As some of the difficulties in the comparison between the simulated surface temperature and the ones reconstructed

from borehole come from the inversion procedure, comparing directly the observed profile with the one obtained
using a one-dimensional heat advection and diffusion forward model driven with climate model can provide new
insight. This approach is an example of the application of Proxy System Models (PSM) that reproduce directly
processes responsible for the signal recorded in the archive (Evans et al., 2013). PSMs have been applied recently for
several proxies, such as tree ring width (Evans et al., 2013) or water-isotope in ice cores, corals, tree ring cellulose, and
speleothem calcite (Dee et al., 2014). The use of climate model outputs to drive a borehole temperature forward model
has demonstrated the strong coupling between near-surface air and ground temperature changes over decades to
centuries (e.g. Beltrami et al., 2005; García-García et al., 2016; González-Rouco et al., 2003, 2006), and validated
climate model outputs (e.g. Beltrami et al., 2006; Stevens et al., 2008).

Nevertheless, using a PSM also introduces some uncertainties that must be taken into account. A critical point for

borehole temperature is the potential influence of long-term climate changes, such as glacial to inter-glacial cycles,
that is difficult to estimate (Orsi et al., 2012, Rath et al., 2012). In addition, the simulated subsurface temperature
profiles are sensitive to model parameters and inputs, such as snow accumulation, ice thickness, geothermal heat flow
and the physical properties of ice or ground, which may have significant uncertainties.

Previous studies using forward models driven by climate model outputs were focused on ground temperature and

not to borehole obtained in the ice. Here, we will fill this gap by simulating directly subsurface temperature for the
publicly available borehole profiles covering the past centuries in Antarctica, using the one-dimensional heat
advection and diffusion forward model of Orsi et al.(2012). Our goal here is to provide a protocol for evaluating the
climate model ability to reproduce observed low-frequency (multi-decadal to centennial scale) variability. We will
analyze two model-data comparison methods to identify the potential advantage and drawbacks of each approach. The
easiest way is in to directly compare the surface temperature reconstructed from the borehole measurements with the
surface temperature time series simulated by the climate model at the grid point closest to each site. The second way is
to compare the simulated subsurface borehole temperature with the measurement by driving the forward model with
climate model outputs. In this case, we analyze the temperature at a fixed time (the one when observation where taken)
as a function of depth. For simplicity, we will later refer to those two methods as a comparison in the time domain and
depth domain, respectively.





This study is organized as follows. The borehole temperature observations, the climate model results, the forward
model and the sensitivity of the results to key parameters of the forward model are briefly described in Section 2.
Section 3 presents the comparison of simulated and reconstructed surface air temperatures, and the comparisons of
simulated and observed subsurface temperature profiles. Some metrics of Antarctic climate for model validation are
proposed and discussed in Section 4. Conclusions are given in Section 5.
**2.    Data and Methods**
**2.1 Borehole temperature observations and reconstructed surface temperature**
The data used in this study includes measured temperature in four boreholes in Antarctica. We refer to them as
'WAIS', 'Larissa', 'Mill Island', and 'Styx' respectively. Figure 1 and Table 1 provide their locations and
corresponding references. The borehole temperature profiles were sampled in January 2008 and January 2009
(WAIS), December-February 2009/10 (Larissa), the summer of 2009/10 (Mill Island), and the summer of 2014/15
(Styx). As shown in Fig. 1 (in red rectangles), the borehole temperatures are affected by the seasonal cycle in the upper
15 meters (Bodri, et al., 2011, Chap. 1), which is not adequate for the reconstruction of annual mean surface
temperature. Consequently, only the depth under 15 meters is used to reconstruct the surface temperature history and
to compare with simulated subsurface temperature profiles.
**Table 1.** Location of the four boreholes. Elevation is in meters above sea level (m a.s.l.)

| Region | Referenced Name | Latitude | Longitude | Elevation (m a.s.l) | Reference |
|---|---|---|---|---|---|
| West Antarctica | WAIS | 79°28′S | 112°05′W | 1766 | Orsi et al., 2012 |
| Antarctic Peninsula | Larissa | 66°02′S | 64°04′W | 1975.5 | Zagorodnov et al., 2012 |
| East Antarctic | Mill Island | 65°33′25.84″S | 100°47′11.44″E | 503 | Roberts et al., 2013 |
| Western Coast of the Ross Sea | Styx | 73°51.10′S | 163°41.22″E | 1623 | Yang et al., 2018 |



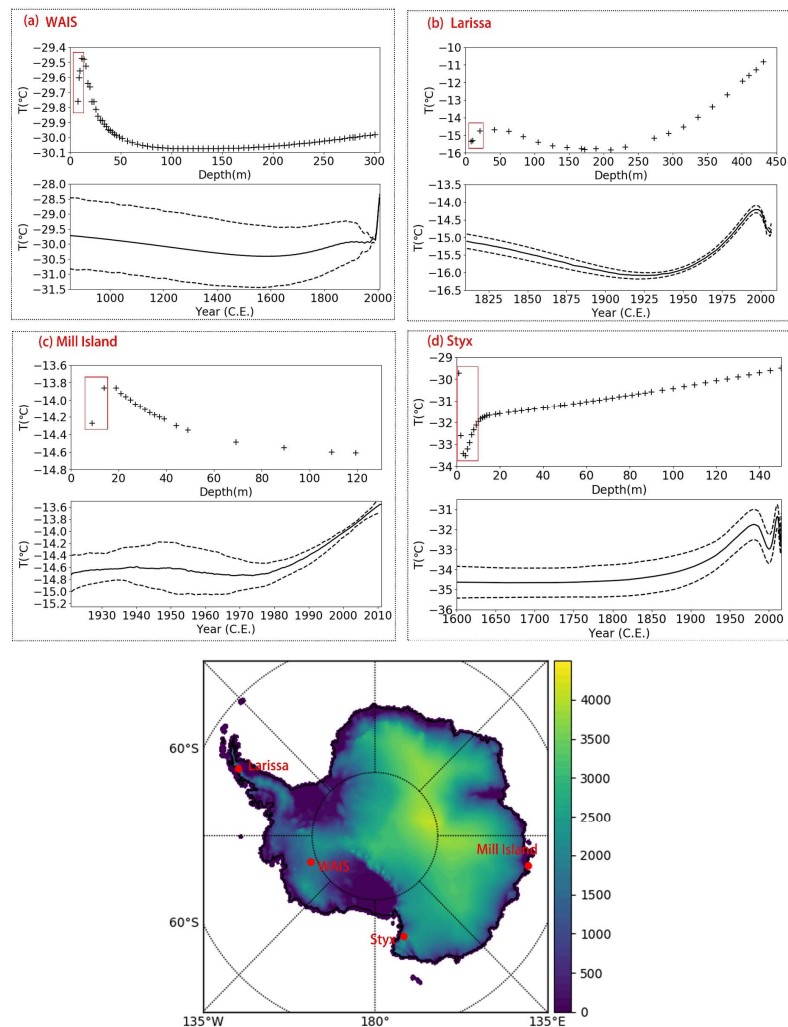

**Figure 1**. Observed Borehole profiles, corresponding reconstructed surface temperature and the location at four sites in Antarctica. The symbols (+) show the measured borehole temperature. The dashed lines represent the reconstructed uncertainty and the thick black line is the mean reconstructed temperature. In (a), (b), (c), and (d), the red rectangles represent the borehole temperatures that are influenced by the seasonal cycle. The bottom panel shows the location of these four boreholes and their corresponding elevation over Antarctica.

The temperature reconstructions and uncertainty estimates for the four boreholes are shown on Fig. 1. For WAIS, Styx, and Mill island, the reconstructed surface temperature series (Fig. 1 a, c, d) are computed using a generalized least-squares algorithm (e.g. Orsi et al., 2012). For Larissa, the surface temperature is recovered by the Tikhonov regularization algorithm (Zagorodnov et al., 2012). This method has been proved to be valid for inverse problems such as the reconstructions based on borehole temperature observations, and the details of this method are explained in (Nagornov et al., 2001, 2006).



**2.2 Climate model simulations**
The surface air temperature used in this study is extracted from PMIP3-CMIP5 experiments (Braconnot et al,2012,
http://pmip3.lsce.ipsl.fr/; Taylor et al., 2012, http://cmip-pcmdi. llnl.gov/cmip5/). Table 2 shows the characteristics
and the corresponding references. These simulations both cover the past 1000 (850-1850 AD) and the historical period
(1850-2005 AD). CESM1-CAM5 and MPI-ESM-P are not continuous in 1850. Such discontinuity for the variables
employed in 1850 falls within the range of variability of the simulated climate, thus merging it with the historical
period have limited effect on the results (Klein et al. 2016). These simulations are driven by natural (orbital, solar
irradiance, volcanic) and the anthropogenic (well-mixed greenhouse gases, ozone, aerosols, land use/land cover)
forcings (Schmidt et al., 2011, 2012). Note that, BCC-CSM1-1 and IPSL-CM5A-LR ignore the impact of land
use/land cover, and IPSL-CM5A-LR does not consider any variations in aerosols and tropospheric ozone. Further
description of the simulations and the forcing can be found for instance in Klein et al., (2016). For CESM1, an
ensemble of simulations is available, providing an estimate of the internal variability as simulated by this model.
**Table 2** Climate model simulations used to drive the forward model

| Name | Model resolution (lat × lon) | Number of simulations for 850-1850 period | Number of simulations for 1850-2005 period | Reference |
|---|---|---|---|---|
| CESM1-CAM5 | 96 × 144 | 12 | 12 | Otto-Bliesner et al., (2016) |
| GISS-E2-R | 90 × 144 | 1 | 1 | Schmidt et al., (2014) |
| IPSL-CM5A-LR | 96 × 96 | 1 | 1 | Dufresne et al. (2013) |
| MPI-ESM-P | 96 × 192 | 1 | 1 | Stevens et al., (2013) |
| CCSM4 | 192 × 288 | 1 | 1 | Gent et al. (2011) |
| BCC-CSM1-1 | 64 × 128 | 1 | 1 | Wu et al., (2014) |

**2.3 The Forward Model Description**
The forward model used herein to simulate the propagation of the signal coming from the surface temperature history
into the subsurface is based on the one-dimensional heat and ice flow equation (Alley and Koci, 1990):
$$\rho c_p \frac{\partial T}{\partial t} = \frac{\partial}{\partial z}\left(k\frac{\partial T}{\partial z}\right) - \rho c_p w\,\frac{\partial T}{\partial t} + Q \qquad (1)$$

where T is the temperature, t is the time, $c_p$ is the heat capacity, $\rho$ is the density of firn/ice, $z$ is the depth, $w$ is the
downward velocity of the firn/ice, $Q$ is the heat production term. The term on the left side represents the change in
heat content and the right terms are the rate of temperature change due to conduction, advection and heat production,
respectively. Important model parameters are summarized in Table 3.





**Table 3** Optimal parameters used to simulate subsurface temperature profile in the forward model driven by the reconstruction for
each site: (a) WAIS; (b) Larissa; (c) Mill Island; (d) Styx

| Site | Surface temperature for steady state (℃) | Accumulation rate (m/second) | Temperature (T) at bottom (℃) | T gradient at bottom (℃/m) | Ice thickness (m) |
|---|---|---|---|---|---|
| WAIS | -29.73 | $6.97 \times 10^{-9}$ | -4.685 | 0.0256 | 3400 |
| Larissa | -16 | $4.147 \times 10^{-8}$ | -10.2 | -0.04 | 447.73 |
| Mill Island | -14.6 | $4.53 \times 10^{-8}$ | -14.6 | 0 | 500 |
| Styx | -32.5 | $2.6985 \times 10^{-9}$ | -20.5 | 0.022 | 550 |

In the model, the density profile, ice thickness and accumulation rates are derived from onsite measurements
according to the descriptions in the original studies while some parameters, such as heat capacity $c_p$, thermal
diffusivity k and heating term Q, are obtained using classical formulations (Cuffey and Paterson, 2010, Chap. 9). The
basal temperature and heat flux for WAIS, Larissa, and Styx are determined using the lower "undisturbed" sections of
the measured borehole temperature extrapolated to the bottom, and for Mill Island, the heat flux is set to zero,
following the original publication. The vertical discretization of the model is not homogenous. For WAIS, a vertical
step of 1 m for the upper 500 m and up to 25 m for the deepest part, and for other sites where the depth of borehole is
close or less than 500 m, the step is set to 1 m for overall depth.
Before the forward model is driven by the climate model results, it is initialized with a stationary profile, which is
generated after a 20000-year model run with a constant climate history and a realistic seasonal cycle. The mean
surface temperature is set to the recent annual average temperature and the season cycle is determined by simplifying
the average over weather station data following Eq. 2 (Orsi et al., 2012).
$$T(t) = 10(\cos(2\pi t) + 0.3\cos(4\pi t)) \qquad (2)$$



**2.4  Sensitivity of subsurface temperature to model parameters**

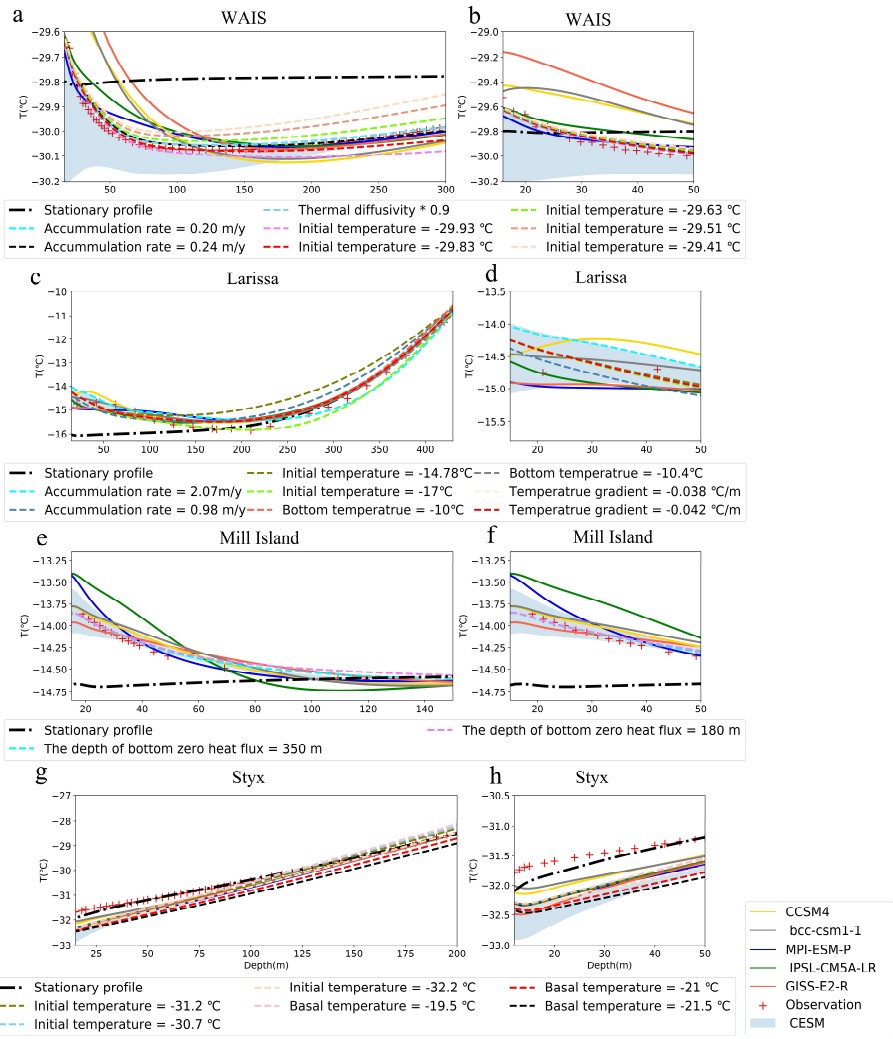

**Figure 2.** Comparison of borehole temperature profiles outputs of 1) GCMs surface temperature time series with optimal parameters (solid lines), and 2) sensitivity tests using the temperature history of once CESM member (dashed lines) at each site. (a) WAIS: 15-300m; (b) WAIS: 15-50m; (c) Larissa: 15-430 m; (d) Larissa:15-50m; (e) Mill Island:15-150m; (f) Mill Island:15-50m; (g) Styx:15-200m; (h) Styx:15-50m. The shade area represents the simulated subsurface temperature ensemble driven by CESM using optimal parameters. The thick dash-dot line denotes the stationary profile at each site.

In order to assess the uncertainty in the model-data comparison related to the parameters of the forward model, it is
necessary to perform a series of sensitivity experiments as shown on Fig. 2. We made different tests for the key
parameters using the values proposed in the original publications (Table 1) and following the protocol of Orsi et al.
(2012).



The range of tested model parameters in the forward model can influence significantly the shape of simulated
subsurface temperature (Fig. 2), which is in good agreement with the previous studies at those sites.
At WAIS-Divide, the spread of the sensitivity tests is lower than the spread if the different scenarios. An increase in
the accumulation rate will reduce the temperature gradient in the borehole profile, but the effect is much weaker than
the difference in temperature histories from the different models. A change in the initial temperature used to calculate
a the starting steady state profile has an influence on the slope of the profile in the deeper part and depth of the
temperature minimum, contributing to the uncertainty in the intensity of the pre-1900 cooling trend and the timing of
the temperature minimum.
At Larissa, the effect of the bottom boundary conditions is important in setting up the temperature gradient from the
bottom to 300 m, and therefore, we will not interpret that segment of the data in terms of climate. It is also evident in
Fig. 2c that the different temperature histories produce a very similar depth profile over that interval.
At Mill island, although the borehole profile is shallow, the ice thickness is much deeper, but unknown. Here we
modeled this by assuming a zero heat flux bottom boundary at various depth. This sensitivity is weak over the data
interval, and the borehole profile is dominated by the surface temperature history.
At Styx, the boundary conditions are adjusted to reproduce the slope of the temperature profile in the deeper part
(100-200m), but the deviation in the top 100 m show that there is climate information stored in the upper part of the
profile, and that this profile cannot be fully determined by boundary conditions.
This confirms that the internal climate variability and different characteristics of these climate models are the
dominant source of the differences and that the model-data comparison provides a robust evaluation of simulated
temperature time series. For the deeper part of WAIS and Larissa, as the shape of subsurface temperature profiles is
influenced by the parameters of the forward model, the evaluation of the long-term cooling trend is more uncertain.



# 3. Results

## 3.1 Comparison between the simulated temperature and reconstructions

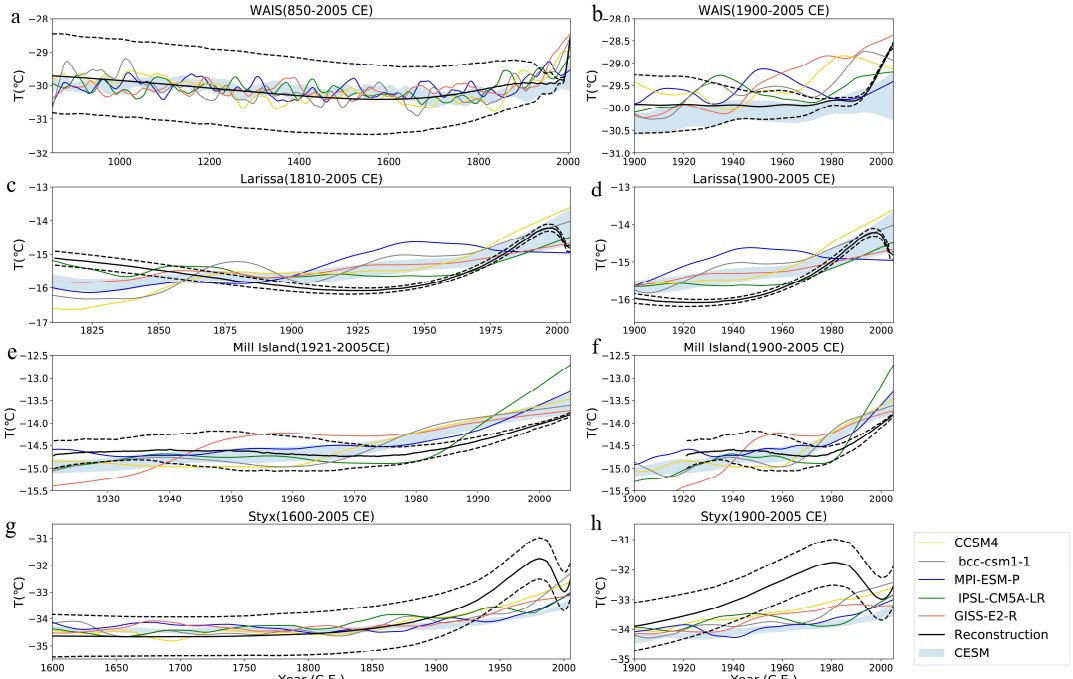

**Figure 3.** Comparison between reconstructed surface temperature series from borehole and the climate model outputs at the grid-point closest to each borehole site. The borehole reconstructions are in black and their uncertainty ranges given by the dashed lines. Color lines correspond to the climate model results. The shaded area represents the mean ±1 standard deviation of CESM model ensemble. For the left column, a 50-year Lowess smoothing has been applied for the WAIS and Styx time series; Larissa and Mill Island are smoothed using 10-year and 3-year windows respectively. The time series in the right column is smoothed using 3-year from 1900 to 2005 C.E.





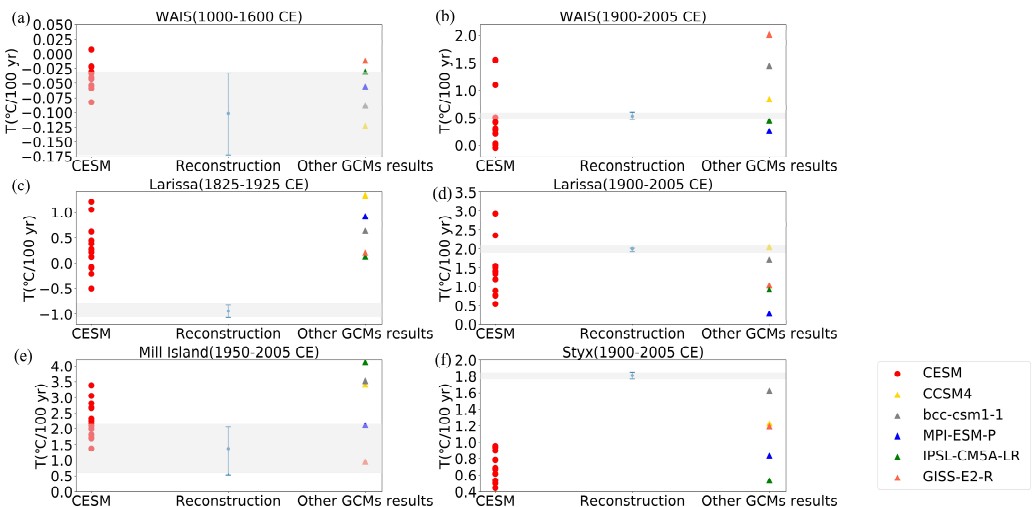

**Figure 4.** Linear trends for the four boreholes over different period: (a) WAIS: 1000 to 1600 C.E.; (b) WAIS: 1900 to 2005 C.E.; (c) Larissa:1825 to 1925 C.E.; (d) Larissa:1900 to 2005 C.E. (e) Mill Island: 1950 to 2005 C.E.; (f) Styx:1900 to 2005 C.E..

Figure 3 displays the comparisons between climate model results and temperature reconstructions from borehole for the four selected sites. In order to remove the bias on the mean state for each climate model, anomalies are shown using the total period covered by each reconstruction as reference. Due to the nature of physical diffusion, the heat propagation acts similarly as a low-pass filter. The reconstructions thus suffer from an attenuation of high frequency temperature variability that becomes stronger as times goes back (Beltrami et al., 2006; Harris and Gosnold, 1999). For instance, in the reconstructed surface temperature of Styx, the point corresponding to 1800 CE in the curve may represent an average temperature between around 1600 CE and 1900 CE while at 1900 CE it corresponds to an average over around 200 years. This characteristic complicates the model-data comparison and trends as shown on Fig. 3 must be interpreted carefully because of this inhomogeneous smoothing.

Because of the internal variability of the system, a single simulation is not expected to reproduce well all the characteristics of the observed variations. The difference can be large, in particular at the local level (e.g. Goosse et al. 2005) but the observations should correspond to a credible member of an ensemble of simulations. Ensuring this compatibility can be achieved using various techniques but the first step is to simply check if the reconstruction is within the range provided by the ensemble (e.g. PAGES2k-PMIP 2015).

Considering the large uncertainty range in these reconstructions, the climate models are visually able to reproduce the general characteristics of reconstructed temperature variability, particularly in the long-term cooling during the last millennium and the recent warming (Fig. 3 and 4). Nevertheless, disagreements have also been identified.

The first major feature in the data is the long-term cooling trend, visible at the WAIS-Divide and Larissa sites. At Larissa (Antarctic Peninsula), the borehole temperature reconstruction finds a cooling trend of -0.94 ± 0.12 °C/century from 1825 to 1925 (Zagorodnov et al., 2012). None of the models are able to reproduce this observation, and instead, they all show a warming trend of comparable magnitude (Fig. 3c and 4c). At WAIS-Divide, the borehole temperature inversion also shows a long-term cooling trend, from 1000 to about 1600 C.E., with a magnitude of -0.10 ±



0.07°C/century (Fig. 3a). The large uncertainty in the long term trend is principally due to the uncertainty in the initial
surface temperature (Fig. 2a; Orsi et al., 2012, their Fig. 3). The quantitative comparison between the trend of
reconstructions and climate model outputs (Fig. 4a) indicates that the simulations generally show a cooling trend over
1000-1600 CE, in agreement with previous studies (e.g., Goosse et al. 2012, Abram et al. 2016, Klein et al. 2019), but
with a lower amplitude, particularly GISS (-0.01°C/century) and IPSL (-0.03°C/century), but most remain within the
lower end of the reconstructed uncertainty range. This long-term cooling trend is a feature of the Antarctic climate that
is visible in many other ice core records (Stenni et al., 2017). A recent compilation of 2K datasets calculated a trend of
-0.26 to -0.4°C/1000 years for the period 0-1900 AD for the Antarctic continental average (Stenni et al., 2017). In the
high latitudes of the Southern Hemisphere, the origin to this millennial-scale cooling is currently not well understood,
but an intermediate complexity model has shown a multi-millennial cooling in summer because of a delayed response
to the decrease in local spring insolation (Renssen et al., 2005) with also a potential influence of volcanic forcing
(Goosse et al. 2012, Abram et al. 2016, Stenni et al. 2017).
A second feature of the data is a warming trend in the twentieth century, which started at different times in the
different records. Styx shows an early warming trend from 1900 to 1980, and a stabilization of the temperature
afterwards (Fig. 3h). Models tend to show the opposite timing, with nearly no trend from 1900 to 1960, and a late
warming trend that differs in amplitude between models. Overall, the warming of the 20th century is about half of
what is observed (Fig. 4f), with bcc (1.63°C/century) and CCSM4 (1.23°C/century) having the largest trends, closest
to the observations (1.81°C/ century).
Larissa shows a temperature minimum in 1940's, followed by a steady warming trend until around 1995. The
magnitude of the 20th century trend is 1.99°C/century. Most model reproduce the timing of the warming reasonably
well, with the exception of MPI, which shows an early warming, but no trend in 1940-2005, and GISS, which has a
very muted trend. If the trend present in the other models is too low, it is rather because of the lack of cooling in the
preceding century, than because of errors in the latest decades.
Mill Island shows a late warming trend starting in the 1980's. Models tend to overestimate this trend (Fig. 4e), in
particular IPSL, bcc and CCSM4. Similarly to Mill Island, WAIS-Divide also shows a positive trend over the period
1900-present that intensifies after 1980. The amplitude of the 20th century warming (0.53 °C/century) is well
simulated, but the start of the trend is often too early, with the exception of CESM, bcc and IPSL, which show a late
warming trend (Fig. 3b).
Overall, the large variability of the trends over the 20th century within the CESM ensemble for WAIS and Larissa
suggests that many apparent model disagreements for those sites can be due to internal variability while the
disagreement may be more profound for Styx and Mill Island.
However, as stated above, borehole temperature reconstructions are "underdetermined", which means that there are
many possible temperature histories that can fit the data. The next step is to determine if the differences between
simulated and reconstructed time series can be discriminated when analyzing observed and simulated temperature
profile

**3.2 Comparison of the simulated subsurface temperature with observation**


The simulated subsurface temperature profile is the results of the superposition of two components: (1) the initial
temperature profile that incorporates the effects of basal heat flux, vertical advection due to ice accumulation and
initial temperature; (2) the subsurface temperature deviations arising from the surface temperature variability. Since
the initial temperature profile for each borehole is obtained by driving the forward model with the optimal parameters
obtained from the original publications describing the reconstructions (see Section 2.4), the differences among the
simulated borehole profiles for each location are caused only by the changes in the upper boundary, i.e. in the climate
model outputs. The simulated subsurface temperature profiles for each borehole are displayed in Fig. 5.

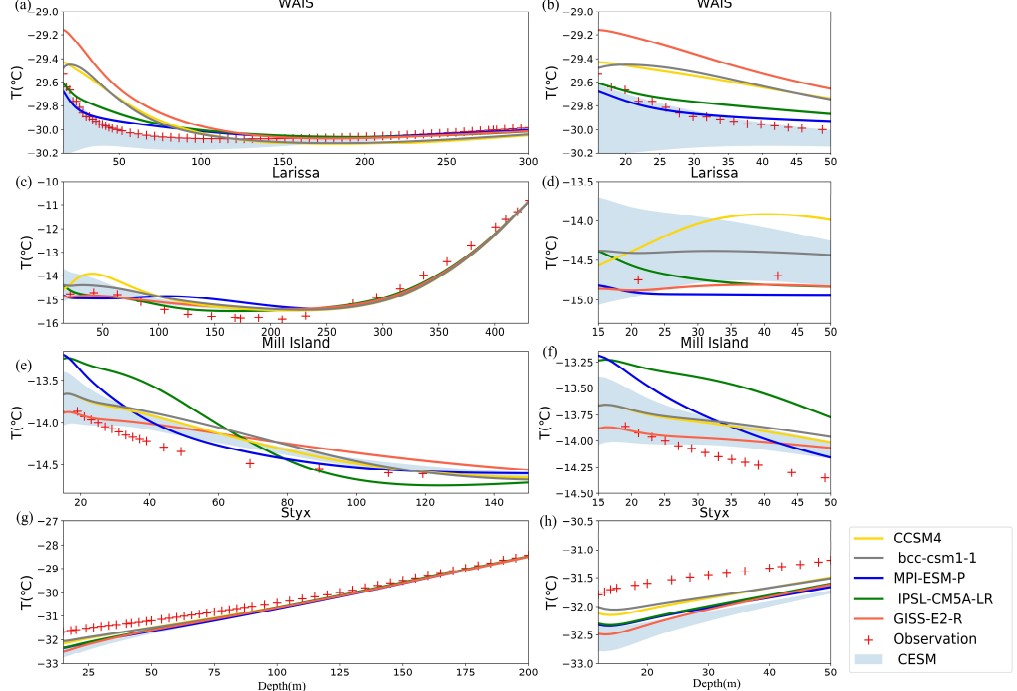


**Figure 5.** Comparisons between simulated subsurface temperature and measurements for : (a) WAIS: 15-300 m; (b) WAIS: 15-50 m; (c) Larissa: 15-430 m; (d) Larissa:15-50 m; (e) Mill Island:15-150 m; (f) Mill Island:15-50 m; (g) Styx:15-200 m; (h) Styx:15-50 m. The shaded area represents the simulated subsurface temperature ensemble driven by CESM ensemble. The right panel is a zoom over the upper 50 m for each borehole.

As previous studies shown (Bodri, Louise, and Vladimir Cermak, 2011, Chap. 2), a 'U' shape subsurface
temperature profile is a direct evidence for the past climate change with a minimum that separates the deeper warming
trend due to geothermal heating and shallower warming trend related to a recent temperature increase (Orsi et al.,
2012; Stevens et al., 2008). Among these four sites, WAIS and Larissa have such characteristics of 'U' shape curve.
For Mill Island, this is less clear but a significant breaking point in each simulated subsurface temperature profile
reflects the surface temperature warming over recent decades while for Styx such break does not seem to be present at
all, but the slope does increase with depth.



Aided by these key properties, we can identify a link between the interpretation in the depth domain and in the time
domain. The analysis of the simulated and observed temperature profile confirm the main conclusion obtained in
section 3.1, in particular with an agreement between model and data on the general tendencies, characterized by a
long-term cooling trend over last millennium and the recent warming. For the deeper part of the profile, the
temperature simulated in experiments driven by MPI, IPSL, GISS at WAIS almost coincides with the corresponding
observations, but they fail to reproduce the depth of the temperature minimum around 120 m in the data. This is
consistent with the fact that IPSL and MPI are at the edge of the reconstructed cooling trend of the last millennium and
GISS present a significant underestimate (Fig. 4a). On the other hand, the CESM ensemble follows the borehole
temperature profile (shaded area on Fig. 5a), and could also reproduce the magnitude of the cooling trend for some of
the members (Fig. 4a). Specifically, the minima in the simulated profiles driven by MPI, IPSL, GISS and CESM
shows the value of -30.06 ℃, -30.06 ℃, -30.07 ℃ and a range of -30.8 ℃ to -30.17 ℃ respectively, which is very
close to the minimum of -30.08 ℃ in the observations. At Larissa, the bottom (270-450 m) of the profile is controlled
by boundary conditions (Fig. 2c), and contains no climate information, as demonstrated by the fact that all curves are
on top of each other on Fig. 5c. Additionally, no simulation has a pronounced inflection point around the 170 m as in
the observation. These characteristics are perfectly consistent with the lack of a cooling trend from mid~19th century
to the early 20th century in the simulations (Fig. 3c). We conclude from this that the cooling trend of 1825-1925 is a
robust feature in the data that can be used to benchmark climate models.
For the recent warming, we see some significant discrepancies among the simulated subsurface temperatures driven
by different climate models at the four boreholes in the depth domain that are consistent with the signal analyzed in the
time domain. For WAIS, in the uppermost part, the simulated subsurface temperature profiles driven by GISS, CCSM
and bcc display significantly higher temperatures than in the observations, while IPSL and MPI-simulated profiles are
close to the measurements (Fig. 5b). This is in perfect agreement with the too high temperatures in models compared
to the reconstructions in the second half of the 20th century (Fig. 3b). For Larissa, all simulated profiles display an
increasing temperature toward the surface as in observations but with different magnitude and shape (Fig. 5c). The
temperature in the simulation driven by the MPI displays a relatively rapid increase until around 100 m and then is
constant, which is consistent with the near constant temperature from 1940-2005 (Fig. 3d). For the ones driven by
CCSM and bcc, they are warmer than the observations between the depth 15m to 50m, which reflects the consistently
warmer temperature shown in Fig. 3d. IPSL-simulated subsurface temperature profile displays the largest similarity to
the observations, whilst the simulations performed with CESM can cover almost all the observation in the shallow
zone. For Mill Island, the simulated subsurface temperature profiles are warmer than observations above 50 m,
confirming the too large warming trend deduced from the analysis of surface temperature. In particular, the IPSL
model has the largest warming trend (Fig. 3 e, f) and also has the warmest temperature profile (Fig. 5 e, f), followed by
MPI. The borehole data thus is providing constraints to evaluate the different simulations. For Styx, the main
discrepancies occur over the shallow depths, between of 15 m to 60 m, where all the simulations depict colder
condition compared with observations (Fig. 5 g, h), as for the surface temperature over the recent decades on Fig. 3.
Nevertheless, we also find in the depth domain some signals that are not obvious in the time domain. In particular, for
WAIS, one of the CESM runs matches the warming trend of the top 100 m, while in time domain the CESM ensemble



was significantly colder than reconstruction over recent decades. The CESM outputs generally follow the data in the
deeper part of the profile (200-300 m), and have an even steeper slope between 100 and 200 m (Fig. 5), while in the
time domain, the cooling trend was underestimated (Fig. 4a). In addition, for WAIS, the simulated subsurface
temperature driven by CCSM4 and bcc over the deeper part of the profile are colder than observations, but start the
warming trend deeper, at about 200 m against 120 m in the observations. This seems puzzling because, in the time
domain, the cooling trend continues until 1800 for CCSM4 (Fig. 2a, yellow), but the larger warming in the last 100
years, is probably shifting the temperature minimum downwards. This example shows that it is difficult to pinpoint the
date corresponding to a temperature minimum in the depth profile, because it depends on the respective speed of
warming and cooling before and after. At Mill Island, in the deeper part (around from 140 m to 100 m) of the profile,
the simulated subsurface temperature profile driven by IPSL, with a slightly decreasing temperature and a colder
climate than observations, is very different from the other ones. However, in the time domain, the difference compared
to other time series for IPSL was much less clear but the consistency between these two domains still exists, and
especially the temperature minimum in 1980 might correspond to the deeper part in the depth domain.
The comparison between the analyses in the two domains appears thus complementary and instructive as it
illustrates that the interpretation may be easier in one case or the other. It also shows that the different model runs
produce different borehole temperature profile, and that the observations can help evaluate the models. In particular,
the analysis of the simulated temperature profile confirms that CESM ensemble can reproduce the multi-decadal and
centennial climate variability at WAIS.

## 4. Proposed metric of Antarctic climate for model validation

In this section, we use the results of the previous section to describe a few metrics that can be used easily to evaluate
the next generation of climate model simulations (e.g. PMIP4-CMIP6, Jungclaus et al., 2017), and investigate the
spatial representativity of the records.

### 4.1 Metric 1 : last millennium cooling at WAIS Divide

Of the four records presented here, WAIS-Divide has the longest retrievable history. We propose here to use the
temperature trend of the period of 1000 to 1600 C.E. as a metric, with the magnitude of -0.102 ± 0.07 ℃/century (Fig.
4a). The end of the cooling trend is not clearly defined by the data, due to the complex time-varying smoothing of the
borehole temperature record, but 1600 C.E. seems to be safely in the cold interval (See Orsi et al., 2012, Fig. 4a for
details). The start of the period is more open, and we chose 1000 C.E. to be compatible with last millennium
simulations. External evidence from a compilation of water isotope records indicates that the cooling trend extended
likely from 0 to 1900 C.E. in many parts of Antarctica (Stenni et al., 2017). It is a robust feature of the Antarctic
climate of the last 2ka, and the WAIS-Divide record is unique in providing a clear quantification of the temperature
trend.
In Fig. 6, we show the 1000 to 1600 C.E. surface temperature trend at WAIS-Divide and at other sites in Antarctica
from the models output. Visually, for most simulations, the cooling at the grid point of WAIS-Divide is similar to the
one obtained at many location in West Antarctica. Only the first member of CESM shows a small warming trend in



West Antarctica. The large spatial coherence of the trend indicates that, although we are making a single point
comparison, it represents a signal common to a large part of the continent. It is also important to estimate the
magnitude of the trend at WAIS compared to other regions. To do so, we calculate the ratio of the trend of surface
temperature from 1000 to 1600 C.E. at any location with the one at WAIS-Divide (Fig. 7). Except the first member of
CESM, if the value is greater than 1 (shown in red tones), it means the trend at the grid-point is larger than that at
WAIS-Divide; if the value lies between 0 and 1 (shown in blue tones), it means the trend at the grid is less than that
observed at WAIS-Divide. Negative values (i.e., a trend of a different sign compared to WAIS) are not shown and the
corresponding region left blank. Since the goal of Fig. 7 is to show the intensity of cooling at WAIS compared with
other points in Antarctica, the first member of CESM 1, which shows a warming trend close to zero at WAIS, is not
very meaningful but it is still included for completeness. For most of the models, WAIS displays a larger cooling from
1000 to 1600 C.E. than other locations in Antarctica (shown in blue) but with magnitude similar to other grid points in
West Antarctica, which is consistent with the reconstruction of Stenni et al. (2017) that shows the largest cooling in
this region over the period 0-1900 CE C.E (Stenni et al., 2017). The spatial patterns of the trends (Fig. 7) are different
between models, but also within the CESM ensembles, showing that the changes in Antarctica are strongly influenced
by internal variability, even at century timescale. Future work including more sites, or using water isotopes and the
Antarctica-2K database will help constrain the spatial pattern of this trend.



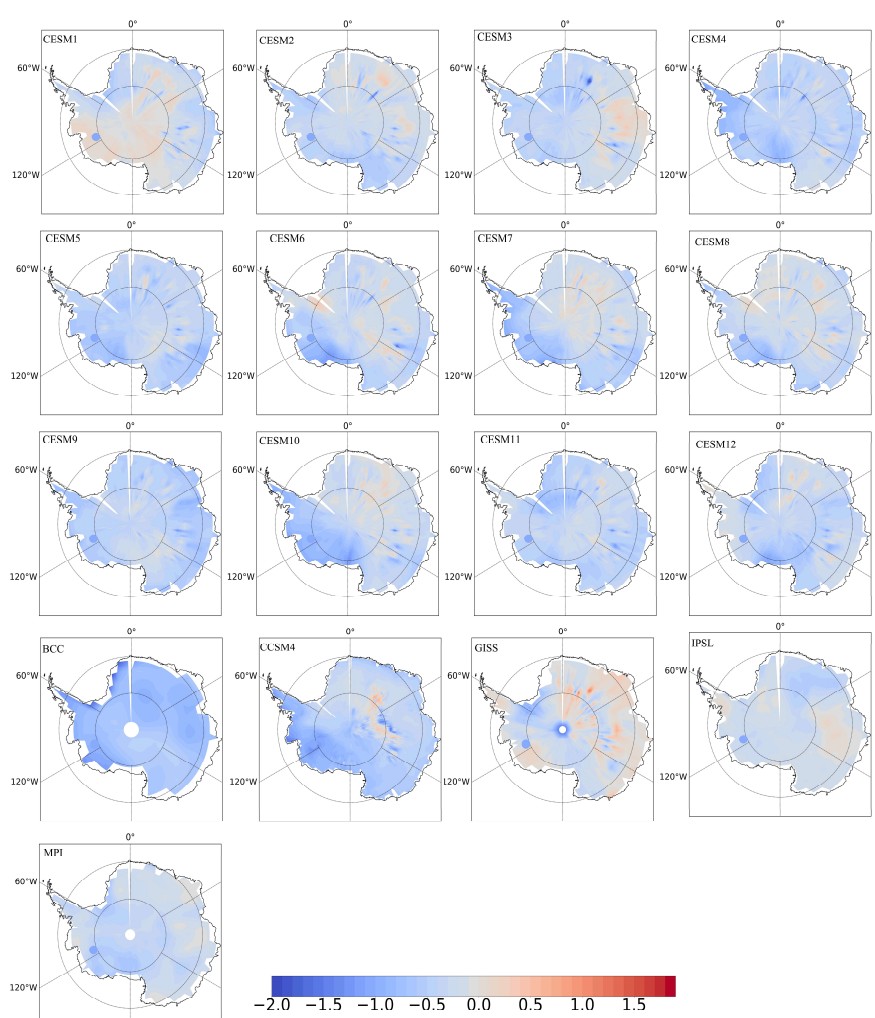


**Figure 6.** The simulated (blue-red shading area) and observed (circle) surface temperature trend from 1000 to 1600 C.E in
Antarctica.



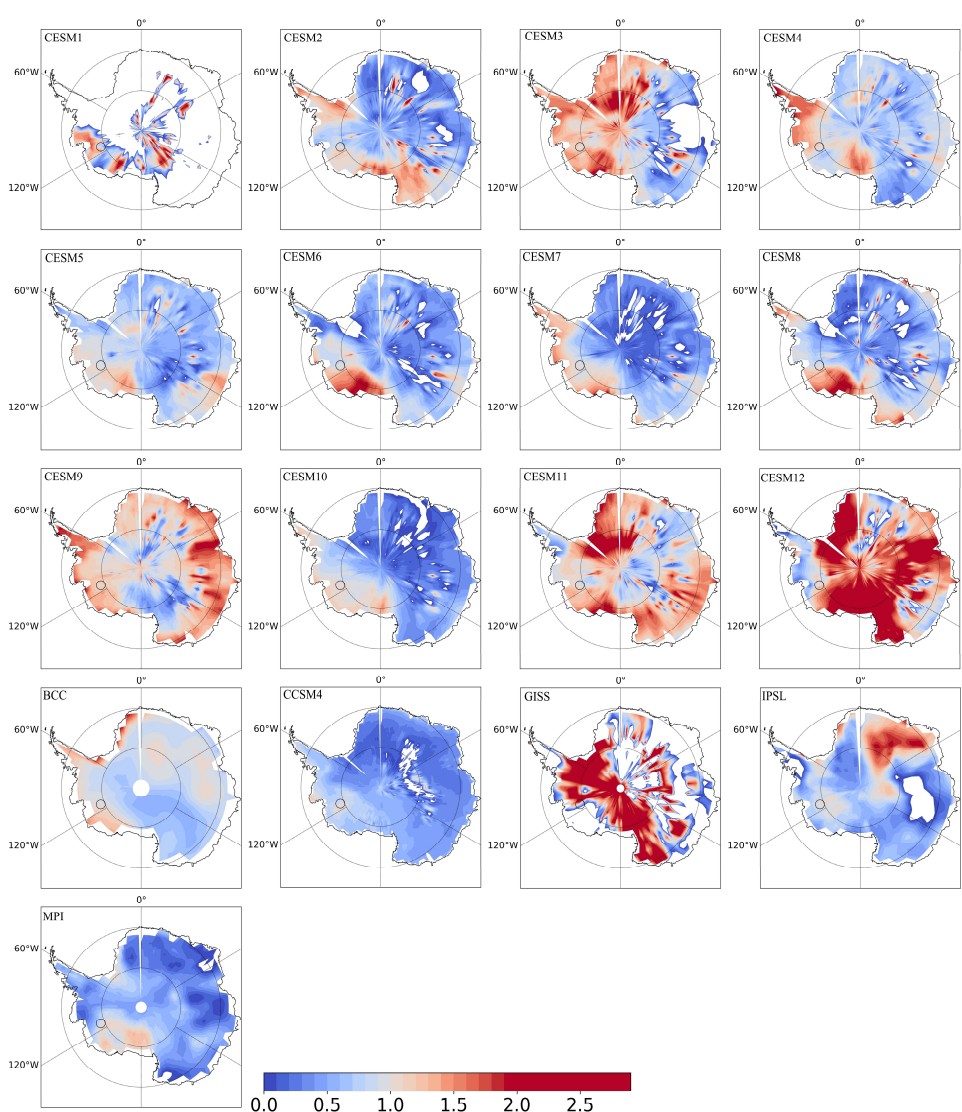

**Figure 7.** The ratio of the surface temperature trend (blue-red shading area) from 1000 to 1600 C.E between other grids in
Antarctica and WAIS-Divide. The black circle denotes the location of the WAIS Divide.
**4.2 Metric 2: nineteenth century cooling at Larissa**
The second metric is the surface temperature trend over the period from 1825 C.E. to 1925 C.E. at Larissa, with the
magnitude of -0.94 ± 0.12 ℃/century. Fig. 8 shows the spatial correlation in the Antarctica Peninsula (AP). Despite
the correlation coefficient decreasing as the grid getting far away from the Larissa, the values, at least around Larissa




for each model, are higher than 0.6, showing that this metric is representative of the whole peninsula region, and not
extremely site-specific.
Figure 9 shows the same temperature trend (1825-1925) for all models. Overall, models are showing a warming
trend (largest for CCSM, MPI and BCC), contradicting the observations, as highlighted already in Fig. 4c. A majority
of the CESM members(CESM1, 7, 8, and 9) show a cooling trend over Antarctica, with CESM 1 and CESM 7 being
able to capture the observed trend.
The 19th century is a time period when the Northern Hemisphere has started warming, whereas Southern
Hemisphere records (Neukom et al., 2014), and specifically Antarctica, show no general warming trend (Stenni et al.,
2017). Models tend to over-estimate the interhemispheric synchroneity (Neukom et al., 2014), and show a warming
trend also in Antarctica , possibly in response to the anthropogenic forcing. This metric is thus an important tool for
future research to evaluate whether the model data mismatch is due to internal variability (which will be investigated
with more ensembles of the same model), or to an overstimated sensitivity to the antropogenic forcing.

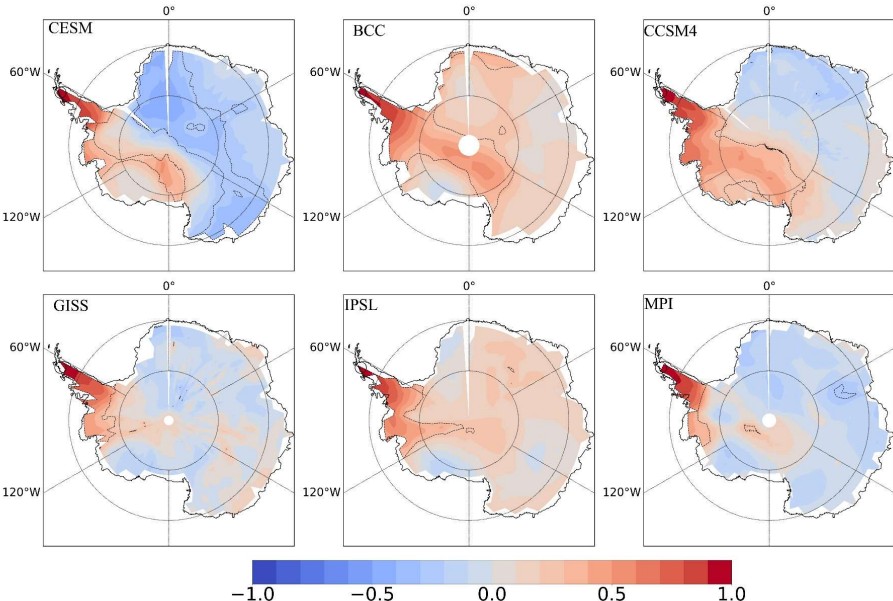


**Figure 8.** The correlation map (blue-red shading area) showing the relationship between the temperature from 1825 C.E. to 1925
C.E at Larissa and other grids in AP for each climate models. The black dotted contour lines show a significant correlation at the 99
% significant level.






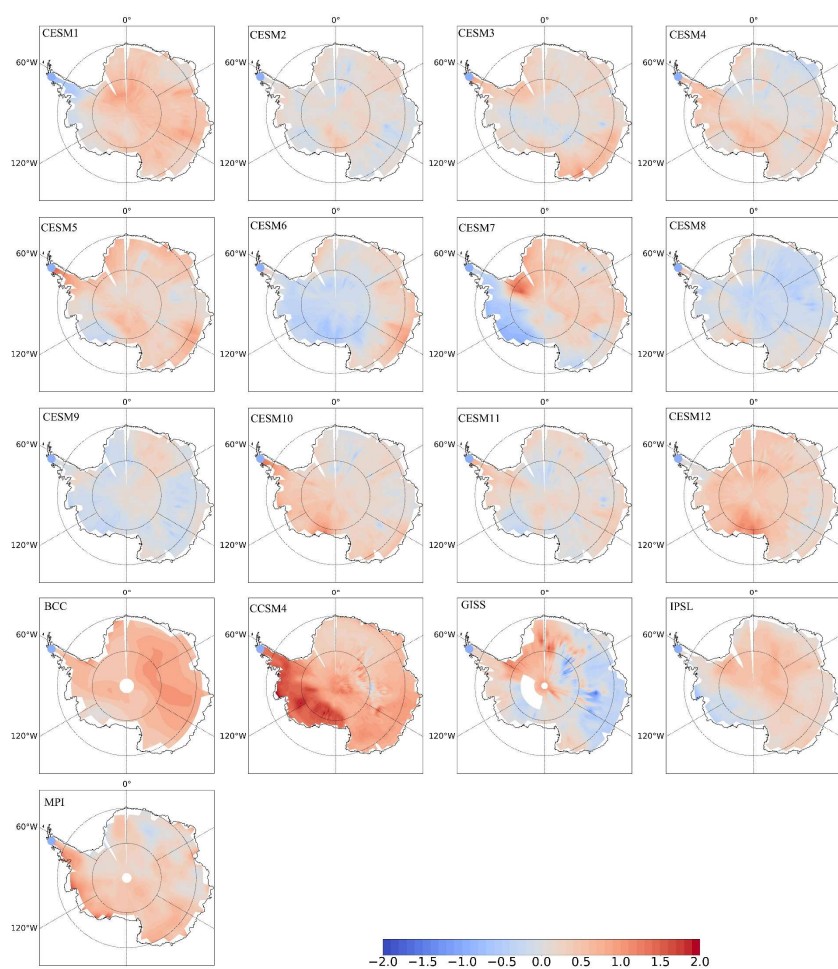


**Figure 9.** The simulated (blue-red shading area) and observed (circle) surface temperature trend from 1825 to 1925 C.E.

## 4.3 Metric 3: recent warming trend

The warming trend of the last 50 years is one of the clearest features of the observations. The intensification of the Southern Annular mode, in response to the Ozone hole is expected to produce a strong warming in the Antarctic Peninsula (incl. Larissa), and cooling on the Antarctic Coast (incl. Styx and Mill Island). WAIS has also been warming significantly over the past decades, and this trend is attributed to variability in the strength and position of the Amundsen Sea low pressure system (Jones et al., 2016). Here we propose a metric of the warming trend from 1950 to 2005 at each of the four sites, to investigate whether model can reproduce these features.

First we look at the spatial correlation of the temperature between each site and other grid points (Fig. 10). The correlation is calculated on annual data for 1950 to 2005 C.E.. It is clear that each of our borehole temperature sites gives information about different sectors of Antarctica. Generally speaking, WAIS is representative of the



West-Antarctic continent, with a more pronounced dipole between WAIS and the Weddell sea section in MPI, and to
a lesser extent CESM and GISS. Larissa is representative of the Antarctic Peninsula as a whole, and from this
resolution of climate model runs, there is no evidence of a dipole between either side of the Transantarctic mountains.
Similar to WAIS, MPI has the strongest expression of a dipole between the Antarctic Peninsula and East Antarctica, a
feature that is weaker but also present in GISS. A model that responds clearly to the Ozone forcing, and has a strong
SAM signature should exhibit this dipole pattern, and it is interesting that some models do not show it, indicating that
the Ozone forcing is not dominating over internal variability. Mill Island is generally representative of the Wilkes
Land sector of East Antarctica, with the largest spatial homogeneity for BCC and IPSL (Fig. 10c). Finally, for Styx,
the models with the largest spatial homogeneity (BCC and IPSL) show a strong correlation between Victoria Land and
the rest of East Antarctica.

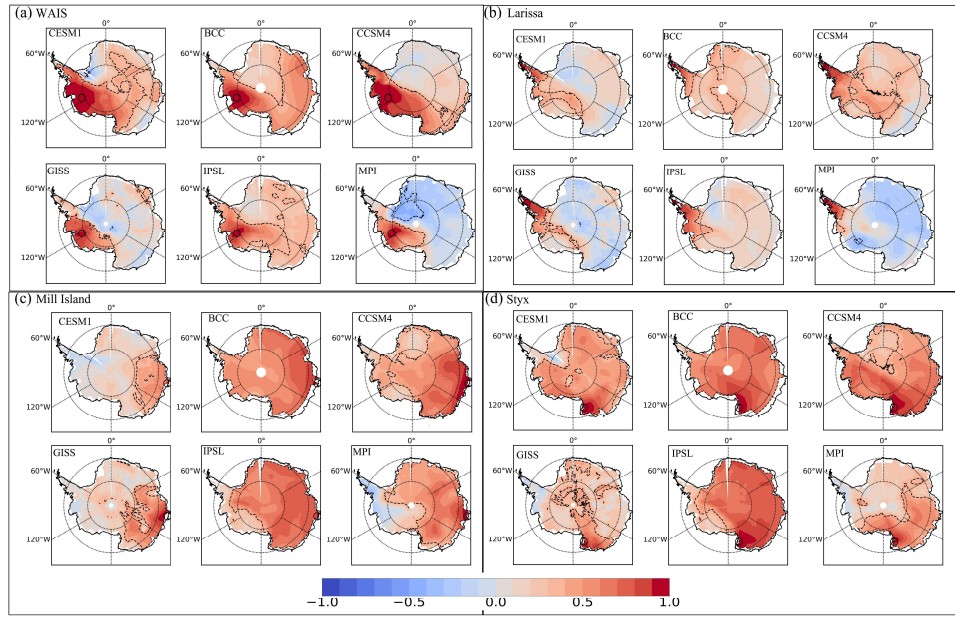


**Figure 10.** The correlation map showing the relationship between the temperature from 1950 C.E. to 2005 C.E at WAIS (a), Larissa
(b), Mill Island (c), Styx (d) and other grids for each climate models. The red dashed contour lines show a significant correlation at
the 99 % significant level.

Figure 11 shows the surface temperature trend from 1950 to 2005 C.E. The strong warming trend at Larissa is
underestimated in most models (Fig. 11 (b)). MPI, which shows a clear dipole between the Peninsula and East
Antarctica (Fig. 10) surprisingly does not show a warming trend at Larissa. This suggests that further work is needed
to diagnose the changes in SAM in those models, and the response of SAM to ozone and greenhouse gas forcing.
Additionally, three out of twelve CESM simulations indicate cooling in West Antarctica, which is coherent with the
hypothesis that the observed warming is due to unforced variability and that models are not expected to match this
trend perfectly. The warming at Mill Island is relatively well reproduced. However, none of the models can reproduce





the weak cooling seen at Styx. The lower spatial representativity of this site (Fig. 10) lead us to interpret this as local
processes missing in low resolution GCMs, such as the correct topography to account for the katabatic wind forcing,
rather than a general failure of models to represent reality.
To sum up, the 1950 to 2005 trend at Larissa of 0.29°C/10-years is a useful benchmark for climate models to test
their response to the Ozone forcing and the temperature pattern associated with the SAM index and other modes of
variability influencing the Peninsula. The trend at Mill Island of 0.14°C/10-years is a useful target to ensure that
Antarctica is not warming too much in response to Greenhouse gas forcing.

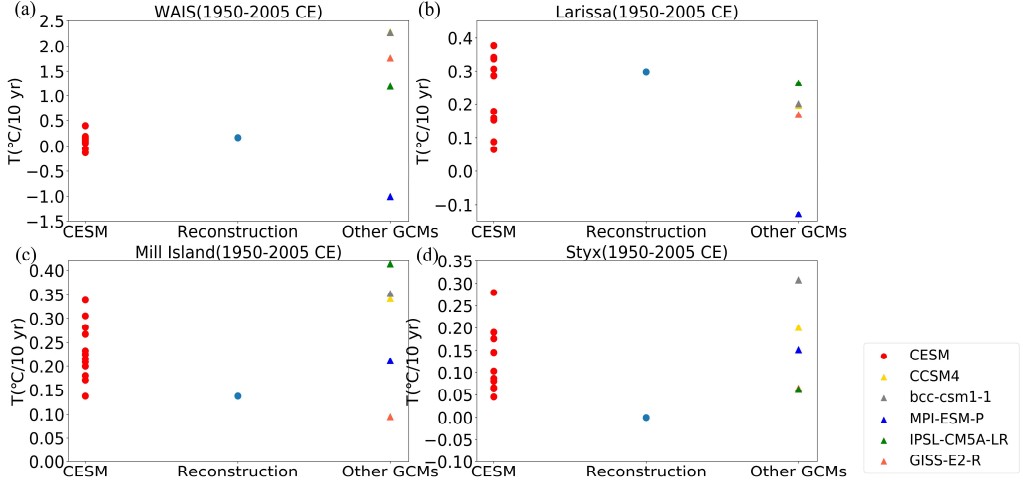

**Figure 11.** Linear trends for the four boreholes over 1950 to 2005 C.E.: (a) WAIS; (b) Larissa; (c) Mill Island; (d) Styx


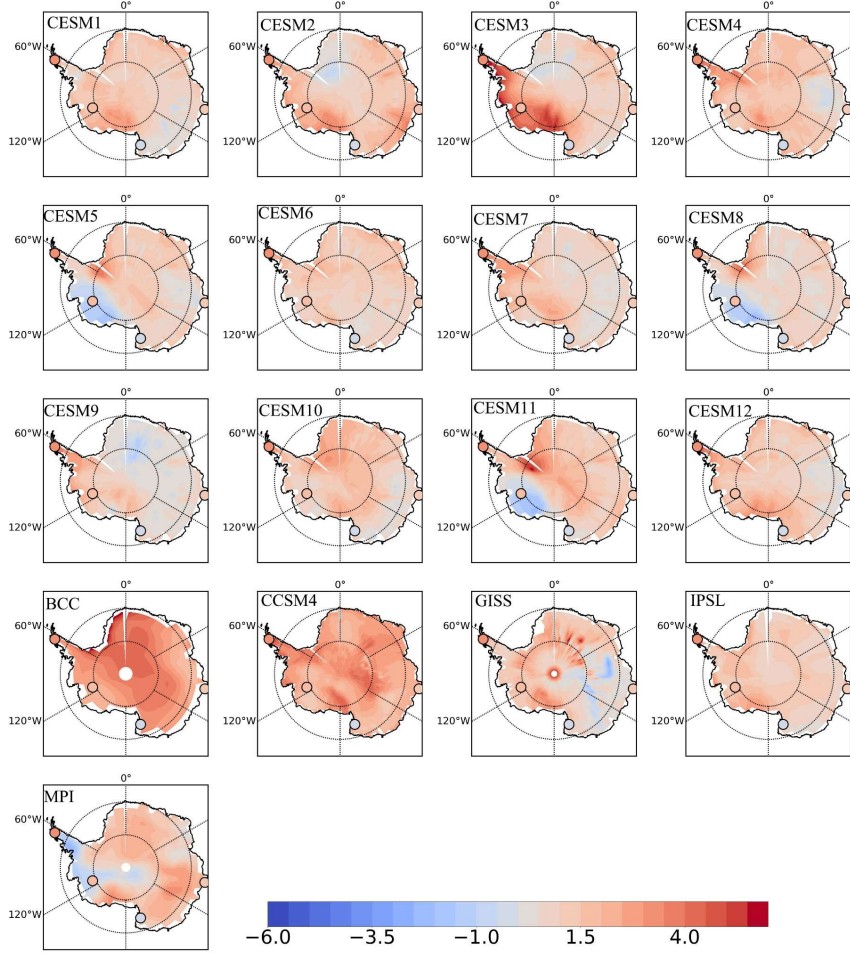


**Figure 12.** The simulated (blue-red shading area) and observed (circle) surface temperature trend from 1950 to 2005 C.E in Antarctica.

## 5. Conclusion

In this study, we test two complementary ways to evaluate the climate model performance using borehole temperature observations. The standard way is to compare the reconstruction of surface temperature with simulated values in the time domain. The successful application here of a forward model driven with climate model results provides an additional way to analyze jointly model results and borehole temperature measurements. Compared to the model-data comparison in the time domain, the forward model allows us to reproduce the subsurface temperature profiles and to compare them directly with measured borehole temperature profiles.



The comparison of the surface temperature time series is simpler and more straightforward but it is limited by the
different resolutions of the reconstructions and climate model results. Nevertheless, some robust conclusions can be
derived from this model-data comparison that are confirmed by the direct analyses of the temperature profiles as a
function of the depth. For instance, the long-term cooling trend over last millennium observed at WAIS is relatively
well reproduced in all models but with a weaker amplitude, which means the model maybe miss some feedbacks or
low-frequency internal variability. Most simulations agree with data on a recent warming but the magnitude and
timing vary a lot between models for the four sites. The large variability of the trends over the 20th century within the
CESM ensemble for WAIS and Larissa suggests that many apparent model disagreements for those sites can be due to
internal variability while the disagreement for Styx and Mill Island may be related to local processes not captured by
global models.
The comparison of the model output and data in the depth domain is useful because the borehole temperature
inversion is an under-determined problem, and many different temperature histories could fit the data equally well.
The comparison of the temperature profiles confirms the conclusions found in the time domain, and validates the
significance of some of the differences found. Some features are however difficult to interpret, such as the depth of the
temperature minimum at the WAIS Divide site, which is not in the same order (deeper = older) as the timing of the
temperature minimum between simulations. This points to the complexity of the interpretation of the borehole
profiles, and the complementary use of the analyses in the depth and time domain.
Finally, some metrics derived from the corresponding reconstructions are proposed to be used more widely in model
evaluation. The metrics used are demonstrated to be representative of a large spatial area, although they are calculated
at a specific site. The results confirm that no models can reproduce the cooling during 19th over the AP and the weak
warming over last 50-years in northern Victoria Land. Nevertheless, these models can capture the larger long-term
cooling from 1000 to 1600 C.E. in West Antarctica, and the recent 50 years warming in West Antarctica and AP. This
work brings quantitative tools to evaluate models and better simulate the Antarctic climate and its response to
forcings.
*Data availability* The PMIP3/CMIP5 model results can be downloaded online from the Program for Climate Model
Diagnosis and Intercomparison (PCMDI; http://pcmdi9.llnl.gov, last access: 20 April 2018). The Forward Model is
available by request to Anais Orsi (anais.orsi@lsce.ipsl.fr).
*Competing interests* The authors declare that there is no conflict of interest.
*Acknowledgements* We acknowledge the World Climate Research Programme Working Group on Coupled
Modelling, which is responsible for CMIP, and we thank the climate modelling groups for producing and making
available their model outputs. This work was supported by the by Chinese Government Scholarships (grant no.
201806040211) and the Belgian Research Action through Interdisciplinary Networks (BRAIN-be) from Belgian
Science Policy Office in the framework of the project "East Antarctic surface mass balance in the Anthropocene:
observations and multiscale modelling (Mass2Ant)" (Contrat n∘ 15 BR/165/A2/Mass2Ant). Hugues Goosse is the
research director within the F.R.S.-FNRS. A. Orsi was supported by the french national programme LEFE/INSU
ABN2K.





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
