# Peer review of "Comparison of observed borehole temperatures in Antarctica with simulations using a forward model driven by climate model outputs covering the past millennium"

_Climate of the Past, 2020_

## Referee Comment (RC1) · Anonymous Referee #1 · 13 Apr 2020

The article provides the first systematic model-data comparison based on borehole temperature-depth profiles in Antarctica. They elaborate two techniques (depth and time domains) to compare these profiles and their reconstructions from four sites with climate model output. They conclude by outlining some useful metrics for future model-data comparison and highlight the importance of internal variability on the observed tendencies.

Below are some points that need clarifying or addressing in the manuscript.

[Figure]

1. L51: "Since the variable measured in the borehole is the temperature itself,..." In most cases, resistivity is measured, which is easily converted to temperature. Is this true of your measurement techniques?

2. L53: "the surface temperature history makes the reconstruction mathematically undetermined." The equation of temperature at depth usually results in a system of linear equations which is mathematically under and overdetermined. Varying mathematical inversion techniques are then utilized to reconstruct the ground surface temperature history. Please clarify. This should also be clarified on L.267.

3. L.92: "Previous studies using forward models driven by climate model outputs were focused on ground temperature and not to borehole..." Please provide a couple examples (references) here.

4. In Section 2.1, the borehole measurements and reconstructions are briefly explained. Since they come from four different publications, how can differences in inversion/reconstruction techniques affect the results presented in Figure 1?

5. L. 136: "CESM1-CAM5 and MPI-ESM-P are not continuous in 1850." What is meant by this? Please clarify.

6. In Section 2.2, for all models, excluding the CESM ensemble, which realization (r1i1p1) is used? Is it the only realization available? If not, why was this one selected? Please clarify.

7. L. 160: "...for Mill Island, the heat flux is set to zero..." How realistic is this? Furthermore, this is a different technique than the other sites. Could this influence the results? If the heat flux is set to zero, how is the steady-state temperature calculated?

8. L. 161: "For WAIS, a vertical step of 1 m for the upper 500 m and up to 25 m for the deepest part, and for other sites where the depth of borehole is close or less than 500 m, the step is set to 1 m for overall depth." Why are various techniques used again? What is the benefit of this?

9. L. 183: "At WAIS-Divide, the spread of the sensitivity tests is lower than the spread if the different scenarios." What is meant by scenarios? Is it the different models being analyzed?

10. L.196: ". . .but the deviation in the top 100 m show that there is climate information stored in the upper part of the profile, and that this profile cannot be fully determined by boundary conditions." Climate is not the sole reason why the top 100 m would show deviation. How can you be sure it is climatic information?

11. At the start of the paragraph at L.198, it is stated that internal climate variability and the different characteristics of the climate models are the main sources of differences. The results from the CESM ensemble have not been discussed in this section. To strengthen this point, I recommend adding in a discussion of it. Furthermore, the statement that internal climate variability and different characteristics of the climate models being the main source of differences does not hold true for Mill Island. In Figure 1, only different depths of the zero heat flux are considered. More tests must be added to conclude the importance of the influence of internal variability and different model characteristics to the differences at this site.

12. In Figure 3, why are different smoothing techniques used? Can it influence the results?

13. The reconstructions from the climate models presented in Figure 3 are calculated using what technique? Their errors bounds are also not presented. How does this influence the results? Does the reconstruction from the climate model always lie within the error bounds? Please clarify in the manuscript.

14. L.215: "In order to remove the bias on the mean state for each climate model, anomalies are shown using the total period covered by each reconstruction as reference." Which figure is being referred to? The paragraph starts discussing Figure 3 but these are not anomalies.

15. Since the temperature variability increases as you go back in time, there is less confidence with respect to the timing of events. Timing of events varies within climate models. Could this further explain any discrepancies of the timing of events? Please discuss.

16. What causes the decrease in temperature at ∼1980 and ∼2000 in Styx and Larissa (Figure 3)? Is it climatic in origin or an artifact of the reconstruction technique?

17. L.390: "Fig. 8 shows the spatial correlation in the Antarctica Peninsula (AP)." Do you mean the spatial correlation of the gridcells? Please clarify.

18. L390-393: "Despite the correlation coefficient decreasing as the grid getting far away from the Larissa, the values, at least around Larissa for each model, are higher than 0.6, showing that this metric is representative of the whole peninsula region, and not extremely site-specific." A correlation coefficient of 0.6 means that it only explains ∼36% of the variance. How can you conclude that it is representative of the entire peninsula?

19. Why are the CESM ensemble members not presented in Figure 8? How is this metric influenced by internal variability?

20. L.426-428: "A model that responds clearly to the Ozone forcing, and has a strong SAM signature should exhibit this dipole pattern, and it is interesting that some models do not show it, indicating that the Ozone forcing is not dominating over internal variability." The CESM ensemble members are not seen in Figure 10 and 11 nor discussed. How can this be concluded?

21. Borehole temperature profiles and their ground surface temperature histories are compared with those from climate models. They ability of the climate models to reconstruct the ground surface temperature was evaluated and three distinct metrics were created. From all of this, how do you think climate models could improve? From your analyses, what are their areas of weaknesses? It would be beneficial to add a section

outlining this to the conclusions.

Technical Points:

There are many grammatical errors throughout the article impeding the reader's comprehension. They are not all outlined below but should be addressed in the revised manuscript.

1. L36: Please define acronym AP.

2. It would facilitate comprehension if the depths of each borehole were added to Table 1. This would help explain the various time periods for the reconstructions found in Figure 1.

3. Please add the units of elevation to the map in Figure 1.

4. In Figure 2, in the boxes below Figures 2a,b and 2c,d, there is a typo in the word accumulation. The thermal diffusion used along with its units should be included. In the caption, " 2) sensitivity tests using the temperature history of once CESM member...", do you mean one CESM member? Also "The shade area represents the simulated subsurface temperature ensemble driven by CESM" should read "The shaded area..."

5. L.192: "At Mill island,..." Should read Mill Island to be consistent throughout the text.

6. L.192: "..the ice thickness is much deeper..." Ice thickness cannot be deeper. Should read thicker.

7. In Figure 4, the y-axis of 4a,e and f are crowded. Either decrease the amount to y-ticks or increase the figure size. Some of the symbols, in particular the yellow triangle of CCSM4, are difficult to see. I would recommend increase the size of the markers for the climate models and the reconstruction. Also, the labels on 4c and d are cut off by the below figures. Please fix.

8. L.254: "Larissa shows a temperature minimum in 1940's..." should read ...1940s.

9. L.270 a period is missing at the end of the sentence.

10. In Figure 5, please use a different colour for the observations.

11. For consistency, use CCSM or CCSM4.

12. For the techniques/metrics elaborated in Section 4, please be consistent with the use of grid, grid-point, and gridcell. Since you are comparing with data from the gridcell, I'd recommend the use of that word to facilitate the reader's comprehension.

13. L.372: "For most of the models,..." It would be best to include a number or percentage of models to really illustrate your point.

14. In Figures 6,9,12, please add the units to the colour bar as these are surface temperatures tendencies.

15. In Figures 6,7,9, the circle illustrating the location of the observations is not clear. Maybe make it bolder or another colour.

16. L.394: "Figure 9 shows the same temperature trend (1825-1925) for all models." Do you mean surface temperature since Figure 9 shows varying trends.

17. L.395-396: "A majority of the CESM members(CESM1, 7, 8, and 9)..." Do you mean minority? 4/12 is not a majority.

18. L.403: There is a typo in the word overestimation.

19. In Figure 8, the dotted contour line is not clear to the reader. Also, indicate that the colour bar represents the correlation coefficient.

20. L.427 please define acronym SAM.

21. Figure 10, some of the numbers in the colour bar appear to be cut-off. The red-dashed line is not visible to the reader. Please correct.

22. Figure 11, the y-label of d is overlapping with c. Please clarify that it is the linear trends of surface temperature in the caption.

23. Figure 12 is not referenced in the text.

---

## Referee Comment (RC2) · Anonymous Referee #2 · 20 Apr 2020

The paper compares temperature observations from 4 Antarctic boreholes with climate model surface temperatures over the last millennium or so. The standard approach to do this is to reconstruct the temperature record from the borehole temperature using a model and compare it with climatic models. The main difficulty is that the thermal diffusivity of ice damps the temperature variations with time and details of the signal are lost: The farther back in time or deeper in the borehole we go, more details are lost. To help the analysis, the authors suggest comparing borehole temperatures with simulated borehole temperatures driven by the climate models using a thermal model. The

authors identify a set of key features in the temperature records from the data: Cooling at WAIS over the last millennium, nineteenth century cooling at Antarctica Peninsula and Antarctic warming over the last 50 years or so. Interestingly, the existing climate models do not reproduce these results. They propose to use these key observations as a metric to test the next generation of climate models.

The paper is in general clear and well written. I have some suggestions for the authors below but I can anticipate that any paper that encourages climate modellers to use data has my full support.

General comments

The borehole distribution is scarce. I know it will always be but I wonder how representative these 4 borehole records are. Inspecting Figure 1, I miss data in the interior of East Antarctica, perhaps Dronning Maud Land; and the coastal area of West Antarctica, Amundsen and Weddle Seas. My view is that a few more sites could improve considerably the benchmark for models.

A detailed description of the climate models, thermal model and borehole temperature data is in other papers. This is understandable but a few short descriptions here and there will improve the clarity of the manuscript considerably. This is of particular importance as the methods used in the manuscript are taping on different scientific areas. To me, for example, Section 2.2 says nothing as I don't know what PMIP3-CMIP5 experiments are, or why they are discontinuous in 1850. I have several suggestions below in the specific comments.

Specific Comments

Title: This is minor point but I think that title is very specific and not easy to digest. What about something like 'Comparing temperature reconstructions from climate models with observed borehole temperature in Antarctica over the last millennium'?

L12-13 In this paper there are two types of 'models': climate and temperature models.

I found data-model confusing here as often papers will compare borehole temperature with modelled temperature. The novelty of this paper is that is comparing 'climate models' temperature with observations. Figure 1. The Temperature vs depth plots don't show the full temperature profile, from surface to bedrock. I assume that the authors are only showing the fraction for the depth that affects the time of interests in the study. This should be made clear. L129 The Tikhonov regularization is a regularization not an inversion method. It doesn't make sense to compare it with the least squares algorithm in Orsi et al 2012.

Section 2.2 I am not a climate modeller, I simply don't understand this paragraph. What are all these acronyms? What is PMIP3-CMIP5 and why are you using the output? What is the discontinuity in 1850? A gentler introduction to the models used in the paper would be welcomed for CP readers.

Section 2.2. Do these models provide surface mass balance as well as temperature? Has the surface accumulation provided by the models been compared with the one observed and used in the temperature model?

Equation 1 I may have missed this but I can't find a description of what is the vertical velocity that the authors are using. I imagine is connected to the surface accumulation but how? How does it vary with depth?

L156 In addition to explaining how accumulation is used in the model, does it vary with time?

L158 The authors are working with shallow temperature, most likely in the firn area. I would like more explanation about how heat capacity and diffusivity depend of density and if density is assumed constant with time.

L158 Heating term in a heat equation is not specific enough. I assume that the authors refer to the internal or strain heating due to flow deformation. How is that calculated? I don't have access to Cuffey and Paterson but I assume that the term depends on the

strain-rates. What components are the authors considering? I am assuming that the term is small but this point requires clarification.

L160 The reasons to apply null heat gradient at Mill Island and explained later and this is confusing. I suggest a clear paragraph describing boundary conditions for Equation 1.

L165-166 How recent is the 'recent annual average'? How does it compare with time-steps?

Equation (2). Is 't' the time in years?

L179 It is not clear to me what this means. Is that 10% variation of boundary and initial conditions? I am assuming that in Larissa the temperature gradient refers to the sensitivity to the bottom boundary condition. Why not in Styx or Mill Island, are they not also frozen to the bed with Neumman boundary conditions? Why some of the sites study more parameters than others? All this should be explained.

L183 'if' should be 'in'

L264-266. I don't understand this paragraph. What is internal variability or a profound disagreement?

Figure 6. I can't see the circles in most of the figures. Perhaps that is good but I would suggest a selection of figures, so that they are bigger or add an edge to the circle.

4.2 Compared with the other sections 'cooling at Larisa' sounds very specific. I suggest 'cooling at AP'.

---

## Author Comment (AC1) · 26 May 2020

**Answer to referee 1**

The referee's comments are shown in black and our answers in blue:

The article provides the first systematic model-data comparison based on borehole temperature-depth profiles in Antarctica. They elaborate two techniques (depth and time domains) to compare these profiles and their reconstructions from four sites with climate model output. They conclude by outlining some useful metrics for future model data comparison and highlight the importance of internal variability on the observed tendencies.

We would like to thank the reviewer for the careful evaluation of our work and the very useful comments that will be addressed in the revised version.

**Specific comments**

1. L51: "Since the variable measured in the borehole is the temperature itself,. . ." In most cases, resistivity is measured, which is easily converted to temperature. Is this true of your measurement techniques?

   Yes, we agree with the referee that resistivity is generally measured in the field, using thermistor, and then it is converted to temperature by Steinhart-Hart equation. But this is true of every modern sensor (an electronic signal is converted to a meaningful variable). We propose to replace the following sentence:

   "Since the variable measured in the borehole is the temperature itself,. . ."

   by :

   "The most significant advantage of borehole paleothermometry is that temperature is directly measured with a thermistor calibrated in the laboratory. Thus, the calibration is independent of the climate at the measurement site."

2. L53: "the surface temperature history makes the reconstruction mathematically undetermined." The equation of temperature at depth usually results in a system of linear equations which is mathematically under and overdetermined. Varying mathematical inversion techniques are then utilized to reconstruct the ground surface temperature history. Please clarify. This should also be clarified on L.267.

   Yes, we propose to replace the following sentence:

   "the surface temperature history makes the reconstruction mathematically undetermined."

   by:

   "Nevertheless, the characteristics of heat conduction that blurs the surface temperature history make the reconstruction mathematically undetermined: several temperature histories can result

in the same borehole temperature profile, because diffusion will smooth out high frequency temperature variations. Consequently, the temperature history cannot be determined unequivocally."

Line 267 reads "However, as stated above, borehole temperature reconstructions are "underdetermined", which means that there are many possible temperature histories that can fit the data." The word "underdetermined" will be between quote with an implicit reference to the explanation given above in line 53.

3. L.92: "Previous studies using forward models driven by climate model outputs were focused on ground temperature and not to borehole. . ." Please provide a couple examples (references) here.

   According to referee's suggestion, we will add some references, Beltrami et al., 2005; García-García et al., 2016; González-Rouco et al., 2003, 2006.

4. In Section 2.1, the borehole measurements and reconstructions are briefly explained. Since they come from four different publications, how can differences in inversion/reconstruction techniques affect the results presented in Figure 1?

   The reviewer raises an important point but comparing the different inversion/reconstruction techniques is out of the scope of our study. The temperature reconstructions are sensitive to the technique used. Notably, because the problem is underdetermined, several temperature histories are equally probable, and the final result will depend on some parameters used to calculate the inversion. We can illustrate this for instance by driving the borehole temperature model selected in this study by the published reconstructed temperature history and compare it to the observed borehole temperature. Difference have been found that are likely attributed to the different methodology and hypothesis but they are relatively small, suggesting that they do not have a major impact on our conclusions. Nevertheless, a more substantial analyses would be required to formally prove this.
   We will add in the revised version a cautionary note mentioning the potential influence of the application of those different techniques.

5. L. 136: "CESM1-CAM5 and MPI-ESM-P are not continuous in 1850." What is meant by this? Please clarify.

   This will be clarified:

   "CESM1-CAM5 and MPI-ESM-P simulations do not cover the entire millennium. Historical simulations covering 1851–2005 C.E. were launched independently of simulations covering 850–1850 C.E. (referred to as the past1000 experiment in CMIP/PMIP nomenclature). In order to obtain results over the full millennium, we adopt the approach from Klein and Goosse (2018) and merge the first ensemble members (r1i1p1) of the past1000 experiment with the corresponding ensemble members of the historical experiment. Although not continuous, there is no large discrepancy in 1850 C.E. between the two merged simulations (e.g., Klein and Goosse, 2018)."

6.  In Section 2.2, for all models, excluding the CESM ensemble, which realization (r1i1p1) is used? Is it the only realization available? If not, why was this one selected? Please clarify.

    Yes, until now only one simulation for CCSM4, GISSE2-R, IPSL-CM5A-LR, MPI-ESM-P and BCC-CSM1-1 is publicly available. This will be clarified in the revised version of the manuscript:

    "For CESM1, an ensemble of simulations is available, providing an estimate of the internal variability as simulated by this model, but for CCSM4, GISSE2-R, IPSL-CM5A-LR, MPI-ESM-P and BCC-CSM1-1, there is only one simulation available."

7.  L. 160: ": : :for Mill Island, the heat flux is set to zero: : :" How realistic is this? Furthermore, this is a different technique than the other sites. Could this influence the results? If the heat flux is set to zero, how is the steady-state temperature calculated?

    In the case of Mill Island, the hole is shallow (120 m), but the ice sheet is very deep at the site. At sites with such a deep ice sheet, and with a high accumulation rate, the conditions at the base are not impacting more than roughly the bottom 1000 m, so it is perfectly reasonable to model the top of the ice sheet only, with a zero heat flux at the bottom. The validity of this assumptions is discussed in detail in the original paper. Here is a quote: "The optimal surface temperature history was found to be essentially independent of the location of this bottom boundary condition for depths in excess of 180 m below the surface" (Roberts et al., 2013). The steady state temperature is calculated in the same way as the other models, and the only difference is the bottom boundary condition. This will be specified in the revised version.

8.  L. 161: "For WAIS, a vertical step of 1 m for the upper 500 m and up to 25 m for the deepest part, and for other sites where the depth of borehole is close or less than 500 m, the step is set to 1 m for overall depth." Why are various techniques used again? What is the benefit of this?

    WAIS is the only very deep borehole, and we use a coarser model resolution for the deepest part to save some computer time as in Orsi et al 2012. This is not required for the shallower cores for which the computation time is lower and it is the reason why we keep a fine resolution for all the depth of the core. This will be specified in the revised version.

9.  L. 183: "At WAIS-Divide, the spread of the sensitivity tests is lower than the spread if the different scenarios." What is meant by scenarios? Is it the different models being analyzed?

    The different scenarios mean the different simulated borehole profiles driven by different climate model results. This will be clarified in the revised version of the manuscript:

    "At WAIS-Divide, the spread of the sensitivity tests is lower than the spread in the simulated borehole profiles driven by different climate model results (solid lines in color in Figure 2 (a) and (b))."

10. L.196: ": : :but the deviation in the top 100 m show that there is climate information stored in the upper part of the profile, and that this profile cannot be fully determined by boundary conditions." Climate is not the sole reason why the top 100 m would show deviation. How can you be sure it is climatic information?

We totally agree that the surface temperature change is not the sole reason why the top 100 m would show deviation. For instance, we can find that the initial and basal temperature have some impacts on the shape of simulated borehole temperature in the top 100 m shown in the Figure 2 (e). Meanwhile, we expect that some climate information is stored in the top 100 m from the comparison between the simulated borehole temperature profiles (solid lines in the Figure 2 ) driven by different GCMs with the stationary temperature profile (thick dash-dot line). This paragraph will be rephrased in the revised version of the manuscript:

"At Styx, the boundary conditions are adjusted to reproduce the slope of the temperature profile in the deeper part (100-200 m). Compared with stationary temperature profile, the simulated borehole profiles driven by GCMs (solid lines in the Figure 4 (e)) show a deviation in the top 100 m, which suggests that there is climate information stored in the upper part of the profile. Meanwhile, at the depth shallower than 50m, the effect of boundary conditions is weaker than the differences in the temperature histories from the different model, which means the borehole temperature data can be used to discriminate between temperature histories provided by the different models."

11. At the start of the paragraph at L.198, it is stated that internal climate variability and the different characteristics of the climate models are the main sources of differences. The results from the CESM ensemble have not been discussed in this section. To strengthen this point, I recommend adding in a discussion of it.

As suggested by the reviewer, a discussion of the internal variability in the CESM ensemble will be added at the start of the paraph at L.198 in the revised version:

"The internal variability also has significant impact on the shape of the simulated borehole profiles. At these four sites, the range of simulations driven by CESM ensemble is much larger than range of the different sensitivity tests in the top of 50 m (shown as the shaded area in Figure 4 b, d, f, h), which conforms that the dominant source of uncertainty in a model–data comparison, at least in the top 50 m, is from the internal variability."

Furthermore, the statement that internal climate variability and different characteristics of the climate models being the main source of differences does not hold true for Mill Island. In Figure 1, only different depths of the zero heat flux are considered. More tests must be added to conclude the importance of the influence of internal variability and different model characteristics to the differences at this site.

Yes, in Figure 2(e), only the upper 50 m shows that there is a noticeable spread between the colored lines, illustrating that different climate model scenarios result in different temperature

profiles, and that this difference is larger than the spread between the dashed lines (the sensitivity to model parameters). We will clarify it and include more sensitivity tests for Mill Island, as requested in the revised version.

12. In Figure 3, why are different smoothing techniques used? Can it influence the results?

The reason why we use different smoothing is to facilitate a comparison between the reconstruction and climate model results. The reconstruction provides a smoothed history of the past surface temperature changes, but the smoothing itself depends on the time and the characteristics of the site. We have tried to mimic this as much as possible by using variable smoothing in the plot. As the reconstructions at WAIS and Styx preserve mainly the centennial and multi-centennial variabilities, we applied longer smoothing (50-year) to the climate model result at WAIS and Styx. Similarly, at Larissa and Mill Island, the reconstructions show the multi-decadal and decadal variabilities, so we choose 10- year smoothing at Larissa and 3-year smoothing at Mill Island.

Using these different smoothing techniques is thus justified and not influencing significantly our conclusions, because our goal here is to perform a visual model-data comparison in the time domain in order to see if the reconstruction is within the range provided by the ensemble. Since the reconstructions have much wider ranges than those ones from the climate model results, the basic compatibility between model and model will not be changed.

13. The reconstructions from the climate models presented in Figure 3 are calculated using what technique? Their errors bounds are also not presented. How does this influence the results? Does the reconstruction from the climate model always lie within the error bounds? Please clarify in the manuscript.

The temperatures displayed in Figure 3 come directly from the surface temperature calculated by the climate model, based on its own dynamics and the forcing applied as discussed in section 2.2. Single time series are available for each model experiment without error bounds but providing an ensemble of experiments gives a range of current state-of-the-art models. This range provides a kind of uncertainty associated with model results but relating this to a precise estimate of the error is unfortunately a complex issues as models are for instance not independent of each other, sharing similar parametrizations, and may have common biases, due in particular to the relatively coarse resolution of climate models (Abramowitz et al., 2019;Knutti et al., 2017;Sanderson et al., 2015).

14. L.215: "In order to remove the bias on the mean state for each climate model, anomalies are shown using the total period covered by each reconstruction as reference." Which figure is being referred to? The paragraph starts discussing Figure 3 but these are not anomalies.

Here, we show an example to explain the methodology. In Figure R1, the original climate model result is shown as the red curve. Its mean over the period 850-2000 C.E. is different from the reconstruction. To remove this bias, we applied a very simple bias correction to climate model results, ensuring that after the adjustment the climate models have the same mean over the

reference period as the reconstruction (Yellow curve in the Figure R1). This will be clarified in the revised version of the manuscript as follows:

"In order to ensure that the climate model results have the same mean over the reference period as the reconstruction, we applied a very simple, constant correction to remove the mean bias of the climate model results as shown on the Figure 3."

[Figure]

Figure R1. Comparison between reconstructed surface temperature series at WAIS and the climate model outputs at the grid cell-point closest to WAIS.

15. Since the temperature variability increases as you go back in time, there is less confidence with respect to the timing of events. Timing of events varies within climate models. Could this further explain any discrepancies of the timing of events? Please discuss.

The timing of the events differs indeed between the simulations if those events are related to internal variability and not caused by a specific forcing. This influence of internal variability can be estimated from the difference between the CESM members as discussed in Section 4. In addition, from the Figure R2, the range of CESM members does not increase back in time. This suggests that the temperature variability between the different members of CESM, and thus the associated uncertainty, does not change a lot over the time.

[Figure]

Figure R2. Temperature variability of the CESM ensemble at the grid cell of WAIS. The black square represents the mean of the CESM ensemble in the corresponding time. Their error bound present the 1 standard deviation (1σ) ranges of the CESM ensemble.

16. What causes the decrease in temperature at 1980 and 2000 in Styx and Larissa (Figure 3)? Is it climatic in origin or an artifact of the reconstruction technique?

    The four borehole reconstructions in the manuscript are from the original papers. In the papers related to the Styx (Yang et al., 2018 ) and Larissa (Zagorodnov et al., 2012), the authors have shown that the reconstructions from borehole are consistent with the weather stations, and ice core isotope-derived records. Consequently, the decrease in temperature in 1980 C.E. and 2000 C.E. at Styx and Larissa likely reflect climatic signals.

17. L.390: "Fig. 8 shows the spatial correlation in the Antarctica Peninsula (AP)." Do you mean the spatial correlation of the gridcells? Please clarify

    As suggested, we rewrite this sentence as follows:

    "Figure 8 shows the spatial correlation between the temperature from 1825 to 1925 C.E. at Larissa and other grids cells for each climate model."

18. L390-393: "Despite the correlation coefficient decreasing as the grid getting far away from the Larissa, the values, at least around Larissa for each model, are higher than 0.6, showing that this metric is representative of the whole peninsula region, and not extremely site-specific." A correlation coefficient of 0.6 means that it only explains 36% of the variance. How can you conclude that it is representative of the entire peninsula?

    Yes, we totally agree with the referee that a correlation coefficient of 0.6 is not that high and our view was probably a bit too optimistic. We have changed this sentence following the suggestion:

    "Despite the correlation coefficient decreasing with the distance from the Larissa, the values, at least around Larissa for each model, are higher than 0.6, showing that this metric is representative of part of the AP region, and not extremely site-specific."

19. Why are the CESM ensemble members not presented in Figure 8? How is this metric influenced by internal variability?

    Figure 8 shows the correlation between the temperature from 1825 to 1925 C.E at Larissa and other grids cells in the Whole Antarctica for each climate model. Since there are no significant differences between each member in CESM ensemble, we just show one member (CESM1) as an example in the manuscript.
    This will be mentioned explicitly in the revised version of the manuscript and we will add the figure shown below (Figure S1) in the supplement:

    "As there are no significant differences between each member in CESM emsemble (see in the Figure S1), only one member CESM1 and other GCMs are present in the Figure 8."

[Figure]

Figure S1. The correlation map (blue-red shading area) showing the relationship between the temperature from 1825 C.E. to 1925 C.E at Larissa and other grid cells in Antarctica for each CESM member. The black dotted contour lines show a significant correlation at the 99 % significant level.

20. L.426-428: "A model that responds clearly to the Ozone forcing, and has a strong SAM signature should exhibit this dipole pattern, and it is interesting that some models do not show it, indicating that the Ozone forcing is not dominating over internal variability." The CESM ensemble members are not seen in Figure 10 and 11 nor discussed. How can this be concluded?

We agree with the reviewer that we jump a bit too quickly on the conclusions and a dedicated analysis should be performed to prove this. In particular, the correlation pattern can be also strongly influenced by the spatial response of each model to the ozone forcing itself. To avoid a long discussion on a point not central to our analysis, we have preferred to remove the part of the sentence relating to the respective role of ozone forcing and internal variability.

21. Borehole temperature profiles and their ground surface temperature histories are compared with those from climate models. They ability of the climate models to reconstruct the ground surface temperature was evaluated and three distinct metrics were created. From all of this, how do you think climate models could improve? From your analyses, what are their areas of weaknesses? It would be beneficial to add a section outlining this to the conclusions.

We agree with the reviewer that this is a very interesting issue. Our goal was to provide a test to estimate the performance of climate models. This is of course the basis for model improvements but the link between a bias in one diagnostic and a model improvement is not straightforward and can be very different for different models. Making specific suggestions would require many additional diagnostics, comparisons and tests. Without that, we fear that any additional material we could add on this subject would be too general or speculative to be really informative for the reader. This is the reason we do not plan to develop this in the revised version.

**Technical Points:** There are many grammatical errors throughout the article impeding the reader's comprehension. They are not all outlined below but should be addressed in the revised manuscript.

Thank you for noticing all the following errors that will be corrected. We will check all the details in the grammar and improve it in the revised manuscript.

1. L36: Please define acronym AP

    Thanks, we will add the definition of AP (Antarctic Peninsula).

2. It would facilitate comprehension if the depths of each borehole were added to Table 1. This would help explain the various time periods for the reconstructions found in Figure 1.

    We will add a column for depth in the Table 1.

3. Please add the units of elevation to the map in Figure 1.

    We will add them.

4. In Figure 2, in the boxes below Figures 2 a,b and 2c,d, there is a typo in the word accumulation. The thermal diffusion used along with its units should be included. In the caption, " 2) sensitivity tests using the temperature history of once CESM member. . .", do you mean one CESM member? Also "The shade area represents the simulated subsurface temperature ensemble driven by CESM" should read "The shaded area. . ."

    As suggested, we will add the unit for the thermal diffusivity.
    We will modify "once CESM member" to "one CESM member" and also correct the phraseology of "shade" to shaded.

5. L.192: "At Mill island,. . ." Should read Mill Island to be consistent throughout the text.

    As suggested, we will correct it.

6. L.192: "..the ice thickness is much deeper. . ." Ice thickness cannot be deeper. Should read thicker.

    This will be modified.

7. In Figure 4, the y-axis of 4a,e and f are crowded. Either decrease the amount to y-ticks or increase the figure size. Some of the symbols, in particular the yellow triangle of CCSM4, are difficult to see. I would recommend increase the size of the markers for the climate models and the reconstruction. Also, the labels on 4c and d are cut off by the below figures. Please fix.

    We will fix these problems in the Figure 4, and it will be updated with a clearer figure in the revised version.

8. L.254: "Larissa shows a temperature minimum in 1940's. . ." should read . . .1940s.

*This will be modified as suggested.*

9.  L.270 a period is missing at the end of the sentence.

    *This will be modified as suggested.*

10. In Figure 5, please use a different colour for the observations.

    *We will modify the Figure 5 as suggested.*

11. For consistency, use CCSM or CCSM4

    *As suggested, we will check and replace CCSM by CCSM4.*

12. For the techniques/metrics elaborated in Section 4, please be consistent with the use of grid, grid-point, and gridcell. Since you are comparing with data from the gridcell, I'd recommend the use of that word to facilitate the reader's comprehension.

    *Thanks for your recommendation. We have replaced grid and grid-point by grid cell.*

13. L.372: "For most of the models,. . ." It would be best to include a number or percentage of models to really illustrate your point.

    *As suggested, we will add a percentage in the corresponding sentence : "75% models show WAIS displays a larger cooling from 1000 to 1600 C.E. than other locations in Antarctica (shown in blue) but with magnitude similar to other grid cells in West Antarctica.".*

14. In Figures 6,9,12, please add the units to the colour bar as these are surface temperatures tendencies

    *We will add the units to the colour bar in Figure 6, 9, 12.*

15. In Figures 6,7,9, the circle illustrating the location of the observations is not clear. Maybe make it bolder or another colour.

    *Done.*

16. L.394: "Figure 9 shows the same temperature trend (1825-1925) for all models." Do you mean surface temperature since Figure 9 shows varying trends.

    *Thanks for your remark. The previous sentence has been modified: no model is able to capture the observed temperature trend from 1825 C.E. to 1925 C.E..*

17. L.395-396: "A majority of the CESM members(CESM1, 7, 8, and 9). . ." Do you mean minority? 4/12 is not a majority.

Yes, we made a mistake here. Thank you for that. We propose to replace "A majority of the CESM members(CESM1, 7, 8, and 9)" by "Only four member of CESM (CESM1, 7, 8, and 9) show a cooling trend over AP, but the magnitudes of them are still less than the observed one".

18. L.403: There is a typo in the word overestimation.

Corrected.

19. In Figure 8, the dotted contour line is not clear to the reader. Also, indicate that the colour bar represents the correlation coefficient.

Done.

20. L.427 please define acronym SAM.

We will add the definition of SAM (Southern Annular Mode).

21. Figure 10, some of the numbers in the colour bar appear to be cut-off. The red dashed line is not visible to the reader. Please correct.

We will fix these problems in the Figure 10, and it will be updated with a clearer figure.

22. Figure 11, the y-label of d is overlapping with c. Please clarify that it is the linear trends of surface temperature in the caption.

We will fix these problems in the Figure 11.

23. Figure 12 is not referenced in the text

Yes, we made a mistake here. We will remove it.

Reference:

Abramowitz, G., Herger, N., Gutmann, E., Hammerling, D., Knutti, R., Leduc, M., Lorenz, R., Pincus, R., and Schmidt, G. A.: ESD Reviews: Model dependence in multi-model climate ensembles: weighting, sub-selection and out-of-sample testing, Earth Syst. Dynam., https://10, 91-105, 10.5194/esd-10-91-2019, 2019.

Klein, F. and Goosse, H.: Reconstructing East African rainfall and Indian Ocean sea surface temperatures over the last centuries using data assimilation, Climate Dynamics, 50, 3909–3929, https://doi.org/10.1007/s00382-017-3853-0, 2018.

Knutti, R., Sedláček, J., Sanderson, B. M., Lorenz, R., Fischer, E. M., and Eyring, V.: A climate model projection weighting scheme accounting for performance and interdependence, Geophys. Res. Lett., https://10.1002/2016gl072012, 2017.

Orsi, A. J., Cornuelle, B. D. and Severinghaus, J. P.: Little Ice Age cold interval in West Antarctica: Evidence from borehole temperature at the West Antarctic Ice Sheet (WAIS) Divide, Geophys Res Lett, 39(9), 1–7, doi:10.1029/2012GL051260, 2012.

Sanderson, B. M., Knutti, R., and Caldwell, P.: A Representative Democracy to Reduce Interdependency in a Multimodel Ensemble, J. Clim., 28, 5171-5194, https://10.1175/jcli-d-14-00362.1, 2015.

Yang, J. W., Han, Y., Orsi, A. J., Kim, S. J., Han, H., Ryu, Y., Jang, Y., Moon, J., Choi, T., Hur, S. Do and Ahn, J.: Surface Temperature in Twentieth Century at the Styx Glacier, Northern Victoria Land, Antarctica, From Borehole Thermometry, Geophys Res Lett, 45(18), 9834–9842, doi:10.1029/2018GL078770, 2018.

Zagorodnov, V., Nagornov, O., Scambos, T. A., Muto, A., Mosley-Thompson, E., Pettit, E. C. and Tyuflin, S.: Borehole temperatures reveal details of 20 th century warming at Bruce Plateau, Antarctic Peninsula, Cryosphere, 6(3), 675–686, doi:10.5194/tc-6-675-2012, 2012.

---

## Author Comment (AC2) · 26 May 2020

**Answer to referee 2**

The referee's comments are shown in black and our answers in blue :

The paper compares temperature observations from 4 Antarctic boreholes with climate model surface temperatures over the last millennium or so. The standard approach to do this is to reconstruct the temperature record from the borehole temperature using a model and compare it with climatic models. The main difficulty is that the thermal diffusivity of ice damps the temperature variations with time and details of the signal are lost: The farther back in time or deeper in the borehole we go, more details are lost. To help the analysis, the authors suggest comparing borehole temperatures with simulated borehole temperatures driven by the climate models using a thermal model. The authors identify a set of key features in the temperature records from the data: Cooling at WAIS over the last millennium, nineteenth century cooling at Antarctica Peninsula and Antarctic warming over the last 50 years or so. Interestingly, the existing climate models do not reproduce these results. They propose to use these key observations as a metric to test the next generation of climate models. The paper is in general clear and well written. I have some suggestions for the authors below but I can anticipate that any paper that encourages climate modelers to use data has my full support.

We would like to thank the reviewer for the positive evaluation of our manuscript and the very useful comments that will be addressed in the revised version as detailed below.

General comments

The borehole distribution is scarce. I know it will always be but I wonder how representative these 4 borehole records are. Inspecting Figure 1, I miss data in the interior of East Antarctica, perhaps Dronning Maud Land; and the coastal area of West Antarctica, Amundsen and Weddle Seas. My view is that a few more sites could improve considerably the benchmark for models. A detailed description of the climate models, thermal model and borehole temperature data is in other papers. This is understandable but a few short descriptions here and there will improve the clarity of the manuscript considerably. This is of particular importance as the methods used in the manuscript are taping on different scientific areas. To me, for example, Section 2.2 says nothing as I don't know what PMIP3-CMIP5 experiments are, or why they are discontinuous in 1850. I have several suggestions below in the specific comments.

a) How representative these 4 borehole records are?

When we propose the metrics of Antarctic climate for model validation in Section 4, we display the correlation maps showing the relationship between the temperature at each borehole site and other grid cells. The results illustrate that the metrics are able to be representative of a large spatial area, although they are calculated at a specific site.

b) Inspecting Figure 1, I miss data in the interior of East Antarctica, perhaps Dronning Maud Land; and the coastal area of West Antarctica, Amundsen and Weddle Seas. A few more sites could improve considerably the benchmark for models

We totally agree with the reviewer that a few more sites could improve considerably the benchmark for models. However, the sparsity of the dataset forbids us to evaluate the skill of the climate model results in other parts of Antarctica. We will insist on that point in the revised version of the manuscript.

c) A detailed description of the climate models, thermal model and borehole temperature data is in other papers. This is understandable but a few short descriptions here and there will improve the clarity of the manuscript considerably.

Our goal here is to simulate borehole temperature profiles by driving the forward model with the results from climate models. In order to make our results more robust, we need to consider the uncertainties in the parameters of the forward model. For the boreholes used in the manuscript, the values of the parameters are derived from the original papers describing the data. Those original studies describe the parameters that have the largest impacts on the surface air temperature reconstruction, a reasonable range for those parameters and the associated uncertainties on temperatures. Detailed information is thus provided in those papers but as reviewer suggested, we will add some sentences in the revised version to further introduce how the forward model work. We propose to replace the following sentence:

"The term on the left side represents the change in heat content and the right terms are the rate of temperature change due to conduction, advection and heat production, respectively."

by:

"In the Equation 1, the term on the left side represents the change in heat content. On the right side, the first term corresponds to the rate of temperature change due to conduction based on the Fourier's law. Ice moving vertically (z-direction) with downward velocity, $w$ , carries a heat flux $\rho c_p w$T across a plane of unit area, oriented perpendicular to z, which is accounted for in the heat transfer by advection shown as the second term. The third term, Q, consists of two part: (1) ice deformation (Cuffey and Paterson, 2010, Chap. 9, Eq. 9.30), (2) firn compaction (Cuffey and Paterson, 2010, Chap. 9, Eq. 9.33)."

Besides, we will also suppress the introduction of the parametrization of the forward model in the manuscript, and add the more specific details of the forward model as the supplementary in the revised version the as followed:

"According to the original publications, we applied different methods to fit the density data for each borehole in the model. For WAIS and Styx, the density profiles, $\rho(z)$, were obtained by a quadratic fit to measured bulk density data following Severinghaus et al. (2010). For Larissa, the density profile was approximated following Salamatin (2000). For Mill Island, due to the similarity between the density profiles at Mill Island and Law Dome (van Ommen et al., 1999), the fitting to the density data is described by a piecewise exponential plus linear or dual exponential according to the analysis on the Law Dome ice core density profile (van Ommen et al., 1999). The density is considered to be in a steady state.

For the other parameters in the forward model, the specific heat capacity $c_p$ is calculated by $c_p$ = 152.5+7.122T (J kg$^{-1}$ K$^{-1}$) (Cuffey and Paterson, 2010, Chap. 9, Eq. 9.1, T means the temperature). The thermal conductivity in ice is taken from $K_{ice} = 9.828 \exp(-5.7 \times 10^{-3}T)$ (Wm$^{-1}$K$^{-1}$) (Cuffey and Paterson, 2010, Chap. 9, Eq. 9.2), and the thermal conductivity of the firn is calculated by Schwerdtfeger formula $K_{schwerdtfeger} = \frac{2K_{ice}\rho}{3\rho_{ice}-\rho}$ (Wm$^{-1}$K$^{-1}$) (Cuffey and Paterson, 2010, Chap. 9, Eq. 9.4, $\rho_{ice}$ = 916.5- 0.14 $T$ (kg/m$^3$), $\rho$ is the density of firn/ice). The vertical velocity at the surface is simply the accumulation rate and decreases with depth as the integral of the densification process (compaction) and the strain due to ice flow divergence. The vertical velocity profile is determined by the method of Alley et al. (1990) and Cuffey et al. (1994) with a constant strain rate. For the accumulation rate, we use a constant value derived from their original publication, which is specified in the Table 3 of the main text. The bottom boundary condition is the basal heat flux and basal temperature. The heat flux is determined by matching the slope

of the temperature increase in the bottom section of the record. At Mill Island, this was not possible, because the data do not extend very deep with respect to the total ice thickness. A zero heat flux boundary condition was chosen instead. The validity of this hypothesis is demonstrated in the original study of Roberts et al. (2013). The basal temperature is determined using the lower "undisturbed" sections of the measured borehole temperature extrapolated to the bottom.

In order to save computation time, the vertical discretization of the model is not homogenous. For WAIS, which is the only very deep borehole, a vertical step of 1 m for the upper 500 m and up to 25 m for the deepest part, and for other sites where the depth of borehole is close or less than 500 m, the step is set to 1 m for the whole depth.

Before the forward model is driven by the climate model results, it is initialized with a stationary profile, which is generated after a 20000-year model run with a constant climate history and a realistic seasonal cycle. It is determined from weather station data. At WAIS, it includes a periodic function with annual and semi-annual components, fitted to 3 years of weather station data from WAIS Divide and Byrd station (AMRC, SSEC, UW-Madison) as follows (Orsi et al., 2012):

$$T(t) = 10(\cos(2\pi t) + 0.3\cos(4\pi t)) \text{ (in K)} \qquad \text{(S1)}$$

At Styx, the seasonal cycle is determined by fitting a sinusoidal function to the automated weather station data as follows (Yang et al., 2018):

$$T(t) = 10(\cos(2\pi t) + 0.35\cos(4\pi t)) \text{ (in K)} \qquad \text{(S2)}$$

"

Specific Comments

Title: This is minor point but I think that title is very specific and not easy to digest. What about something like 'Comparing temperature reconstructions from climate models with observed borehole temperature in Antarctica over the last millennium'?

The title is currently "Comparison of observed borehole temperatures in Antarctica with simulations using a forward model driven by climate model outputs covering the past millennium". We consider that one of the main originality of our study is to use a forward model so we prefer to keep this information it in the title.

L12-13 In this paper there are two types of 'models': climate and temperature models. I found data-model confusing here as often papers will compare borehole temperature with modelled temperature. The novelty of this paper is that is comparing 'climate models' temperature with observations. Figure 1. The Temperature vs depth plots don't show the full temperature profile, from surface to bedrock. I assume that the authors are only showing the fraction for the depth that affects the time of interests in the study. This should be made clear.

We use indeed two types of model and this may be confusing. For the revised version, we will check each time 'model' is used and ensure that the meaning is clear.

There is no full temperature profile from surface to bedrock in the Figure 1 because the borehole temperature measurements were made on shallow boreholes that did not always extend through the entire ice sheet. This will be specified in the revised version. Since it is a model-data comparison, we adjust the plot to the depth where there is data. Although it is the fraction of the total depth of the ice sheet, it

is adequate to be used to reconstruct surface air temperature, as done in the literature, and to compare with the simulated borehole temperature profiles obtained by the forward model driven by climate model results at each site.

L129 The Tikhonov regularization is a regularization not an inversion method. It doesn't make sense to compare it with the least squares algorithm in Orsi et al 2012.

Yes, we agree that the Tikhonov regularization belongs to the larger family of least squares regressions. On Figure 1, we reproduce the published datasets and temperature reconstructions. We cannot avoid the fact that records were published with different methods. We don't intend to compare the methods, but we will add in the revised version a cautionary note mentioning the potential influence of the application of those different techniques.

Section 2.2 I am not a climate modeller, I simply don't understand this paragraph. What are all these acronyms? What is PMIP3-CMIP5 and why are you using the output? What is the discontinuity in 1850? A gentler introduction to the models used in the paper would be welcomed for CP readers

We are sorry that we forgot to explicit the acronyms. The full names of PMIP3 and CMIP5 are the third phase of the Past Model Intercomparison Project (PMIP3; Otto-Bliesner et al., 2009) and the fifth phase of the Coupled Model Intercomparison Project (CMIP5; Taylor et al., 2012). The Paleoclimate Modelling Intercomparison Project (PMIP) is a long‑standing initiative that provides coordinated paleoclimate modeling and data collection activities to facilitate advances on the study of the mechanisms of climate change (Otto-Bliesner et al., 2009). The fifth phase of the Coupled Model Intercomparison Project (CMIP5) produced a state-of-the- art multimodel dataset designed to advance our knowledge of climate variability and climate change (Taylor et al., 2012). CMIP and PMIP are major sources of information for the assessment reports of the Intergovernmental Panel on Climate Change (IPCC).

This has been clarified as follows:

"The simulated surface air temperature used in this study is extracted from climate model simulations covering the past millennium performed in the framework of the third phase of the Past Model Intercomparison Project (PMIP3; Otto-Bliesner et al., 2009) and the fifth phase of the Coupled Model Intercomparison Project (CMIP5; Taylor et al., 2012)."

"CESM1-CAM5 and MPI-ESM-P simulations do not cover the entire millennium. Historical simulations covering 1851–2005 C.E. were launched independently of simulations covering 850 - 1850 C.E. (referred to as the past1000 experiment in CMIP/PMIP nomenclature). In order to obtain results over the full millennium, we adopt the approach from Klein and Goosse (2018) and merge the first ensemble members (r1i1p1) of the past1000 experiment with the corresponding ensemble members of the historical experiment. Although not continuous, there is no large discrepancy in 1850 C.E. between the two merged simulations (e.g., Klein and Goosse, 2018)."

Section 2.2. Do these models provide surface mass balance as well as temperature? Has the surface accumulation provided by the models been compared with the one observed and used in the temperature model?

Yes, we can obtain precipitation and sublimation/evaporation from models, to calculate the surface mass balance (SMB) as done for instance in Dalaiden et al (2020). Nevertheless, we did not use simulated SMB

here because our focus is on temperature changes and we did not want that biases in the simulation of SMB influence our conclusions. Consequently, we use the original observed accumulation rate instead of the simulated one. A precise evaluation of the accuracy of the SMB in the climate models and of its impact is the topic of a paper in itself. We will add in the revised version a note mentioning the point as suggested by the reviewer:

"In addition, although we can obtain the simulated surface mass balance (SMB) from these models(e.g. Dalaiden et al., 2020), we do not use it here and keep the observed accumulation rate in the forward model since biases in the simulation of SMB may affect our conclusions and the focus here is on the simulated temperature evolution.

Equation 1 I may have missed this but I can't find a description of what is the vertical velocity that the authors are using. I imagine is connected to the surface accumulation but how? How does it vary with depth?

Yes, we agree with the reviewer that the vertical velocity is connected to the surface accumulation. Vertical velocity depends on the accumulation, the densification process (compaction), and finally the strain due to ice flow divergence. Vertical velocity at the surface is simply the accumulation rate, and it decreases to zero at the bottom, or a constant value equal to the melt rate if there is melting. The detailed vertical velocity profiles for the boreholes are shown in the papers describing the original data. We will add a sentence describing the vertical velocity parametrization in the revised paper.

L156 In addition to explaining how accumulation is used in the model, does it vary with time?

No, we use a constant accumulation rate derived from their original publication, which is specified in the Table 3 of the main text, because the model we use has a constant density profile, and only uses the accumulation rate in the calculation of the vertical velocity. Additionally, sensitivity studies made for WAIS-Divide (Orsi et al., 2012) and Styx (Yang et al., 2018) show that the variations of accumulation are small enough over the period considered that it does not appreciably change the results. We will add a sentence explaining this in the revised paper.

L158 The authors are working with shallow temperature, most likely in the firn area. I would like more explanation about how heat capacity and diffusivity depend of density and if density is assumed constant with time.

The specific heat capacity $c_p$ is calculated by $c_p = 152.5 + 7.122T$ (J kg$^{-1}$ K$^{-1}$) (Cuffey and Paterson, 2010, Chap. 9, Eq. 9.1, $T$ is the temperature). The thermal conductivity in ice is taken from $K_{ice} = 9.828 \exp(-5.7 \times 10^{-3}T)$ (Wm$^{-1}$K$^{-1}$) (Cuffey and Paterson, 2010, Chap. 9, Eq. 9.2), and the thermal conductivity of the firn is calculated by Schwerdtfeger formula $K_{schwerdtfeger} = \frac{2K_{ice}\rho}{3\rho_{ice}-\rho}$ (Wm$^{-1}$K$^{-1}$) (Cuffey and Paterson, 2010, Chap. 9, Eq. 9.4, $\rho_{ice} = 916.5 - 0.14\,T$ (kg/m³), $\rho$ is the density of firn/ice). The density profile is considered to be in steady state.

L158 Heating term in a heat equation is not specific enough. I assume that the authors refer to the internal or strain heating due to flow deformation. How is that calculated? I don't have access to Cuffey and Paterson but I assume that the term depends on the strain-rates. What components are the authors considering? I am assuming that the term is small but this point requires clarification.

The heating term Q consists of two part: (1) ice deformation (Cuffey and Paterson, 2010, Chap. 9, Eq. 9.30), (2) firn compaction (Cuffey and Paterson, 2010, Chap. 9, Eq. 9.33). We will add a section in the supplementary information with these details about the temperature diffusion model.

L160 The reasons to apply null heat gradient at Mill Island and explained later and this is confusing. I suggest a clear paragraph describing boundary conditions for Equation 1.

We suggest rephrasing this section to:

"The bottom boundary condition includes the basal heat flux and basal temperature. The heat flux is determined by matching the slope of the temperature increase in the bottom section of the record. At Mill Island, this was not possible, because the data do not extend very deep with respect to the total ice thickness. A zero heat flux boundary condition was chosen instead. The validity of this hypothesis is demonstrated in the original study of Roberts et al. (2013). The basal temperature is determined using the lower "undisturbed" sections of the measured borehole temperature extrapolated to the bottom."

L165-166 How recent is the 'recent annual average'? How does it compare with timesteps?

The "recent annual average" means the mean temperature that could be derived from weather station data, as described in the rest of the sentence. For instance, at WAIS, it is the average of 3 years of weather station data from WAIS Divide and Byrd station (AMRC, SSEC, UW-Madison). It is larger than the time step which is 200 time steps per year. We will clarify this point in the revised version.

Equation (2). Is 't' the time in years?

Yes, you are right, we will add the description of all the symbols paragraph in the revised version of the manuscript.

L179 It is not clear to me what this means. Is that 10 % variation of boundary and initial conditions? I am assuming that in Larissa the temperature gradient refers to the sensitivity to the bottom boundary condition. Why not in Styx or Mill Island, are they not also frozen to the bed with Neumman boundary conditions? Why some of the sites study more parameters than others?

(a) Is that 10 % variation of boundary and initial conditions?

For the boundary and initial conditions, we followed the tests proposed in the original papers. The 10% sensitivity is only applied for the thermal diffusivity ("*0.9" in the Figure 2a and 2b). Following Orsi et al (2012), we used the Schwerdtfeger formula (Cuffey and Patterson, 2010, Chapter 9), which depends on both temperature and density of the snow. It usually gives an upper estimate of the thermal diffusivity of snow. This is the reason why we decreased the thermal conductivity by 10%, and ran the optimization of the temperature again. Compared with the effect of the initial temperature on the shape of simulated borehole temperature, the thermal conductivity and accumulation rate have no significant effect on the result (Figure R1 ), which is in good agreement with the result in Orsi et al. (2012). Consequently, we prefer to remove the curves of sensitivity tests of thermal conductivity and accumulation rate in the revised version. This will also provide a simpler and clearer description of the remaining sensitivity experiments.

[Figure]

**Figure R1.** The sensitivity tests using the temperature history of one CESM member (dashed lines) at WAIS.

(b) I am assuming that in Larissa the temperature gradient refers to the sensitivity to the bottom boundary condition. Why not in Styx or Mill Island, are they not also frozen to the bed with Neumman boundary conditions? Why some of the sites study more parameters than others?

According to the original papers, the various parameters in the forward model have effects of different magnitude on the results. For instance, at Styx (Yang et al., 2018) and Larissa (Zagorodnov et al., 2012), the bottom temperature has significant influence while the bottom boundary conditions are of limited importance at WAIS (Orsi et al., 2012). Consequently, we test for some sites more parameters than for others. Nevertheless, in the revised version, we will add some sensitivity tests for the initial temperature for the Mill Island, and for the other sites, we will remove those curves which are of limited importance in order to make the Figure 2 clearer. We propose to replace the following sentence:

"In order to assess the uncertainty in the model-data comparison related to the parameters of the forward model, it is necessary to perform a series of sensitivity experiments as shown on Figure 2. We made different tests for the key parameters using the values proposed in the original publications (Table 1) and following the protocol of Orsi et al. (2012)."

by:

"According to the original papers, the various parameters in the forward model have effects of different magnitude on the results. Consequently, in order to assess the uncertainty in the model-data comparison related to the parameters of the forward model, we perform a series of sensitivity experiments on the parameters which have the largest effects on the simulated borehole profiles shown in the Figure 2."

L183 'if' should be 'in'

Corrected.

L264-266. I don't understand this paragraph. What is internal variability or a profound disagreement?

(a) What is internal variability?

Internal variability refers to the climate variability due to process internal to the climate system, in contrast to the forced variability that is a response to forcings like change in insolation, greenhouse gases, etc. Models that run for hundreds of years are not expected to reproduce the timing of the observed variations due to internal variability, in particular the exact phase of internal oscillations, such as the North Atlantic Oscillation, or ENSO. As a result, the same model, with the same physics and the same input forcings can produce different temperature when it is started with slightly different conditions. This is why "ensembles" are run: they are the outputs of the same model, with the same forcings, but slightly different initial conditions. If the results from different ensemble members are different, then this difference is attributable to internal variability rather than to the response to external forcings. By comparing the different temperature histories from the different CESM ensemble members, we can quantify the amplitude of internal variability. In many cases, the model-data difference for the other models is within this range of internal variability deduced from the CESM ensemble. As a result, we hypothesize that the discrepancy between an individual simulation with one model and data is likely due to the poor sampling of internal variability by only one simulation (reality is only one realization among all the possibilities), rather than a problem with model physics or inappropriate forcing factors. This potential role of internal variability will be explained in more detail by rewriting section which was in the submitted version at L264-L266.

(b) What is a profound disagreement?

The lines L264-L266 summarize the main source of model-data disagreement over the 20th century. For Styx (Figure 4(f)) and Mill Island (Figure 4(e)), the discrepancy between all the models and data is larger than the spread of the CESM ensemble. This suggests that the model-data difference cannot be simply attributed to uncertainties associated with the low number of ensemble members, but rather, with a systematic bias, which could come from model physics or input forcings.

This will be clarified in the revised version of the manuscript:

"Overall, for WAIS (Figure 4(b)) and Larissa (Figure 4(d)), the reconstructed trends lie in the CESM ensemble range, suggesting many apparent model disagreements for those sites can be due to internal variability. For Styx (Figure 4(f)) and Mill Island (Figure 4(e)), the reconstructed trends are larger than the spread of the CESM ensemble, which means the disagreements are not only due to internal climate variability but are related to a systematic climate model bias in this region."

Figure 6. I can't see the circles in most of the figures. Perhaps that is good but I would suggest a selection of figures, so that they are bigger or add an edge to the circle.

We will fix the problem in the Figure 6, and it will be updated with a clearer figure.

**Reference**

Cuffey, K., and W. Paterson, The Physics of Glaciers, Academic, Amsterdam, 2010.

Orsi, A. J., Cornuelle, B. D. and Severinghaus, J. P.: Little Ice Age cold interval in West Antarctica: Evidence from borehole temperature at the West Antarctic Ice Sheet (WAIS) Divide, Geophys Res Lett, 39(9), 1–7, doi:10.1029/2012GL051260, 2012.

Otto-Bliesner, B. L., Joussaume, S., Braconnot, P., Harrison, S. P., and Abe-Ouchi, A.: Modeling and Data Syntheses of Past Climates: Paleoclimate Modelling Intercomparison Project Phase II Workshop, Estes

Park, Colorado, 15– 19 September 2008, Eos T. Am. Geophys. Un., 90, p. 93, https://doi.org/10.1029/2009EO110013, 2009

Roberts, J. L., Moy, A. D., Van Ommen, T. D., Curran, M. A. J., Worby, A. P., Goodwin, I. D. and Inoue, M.: Borehole temperatures reveal a changed energy budget at Mill Island, East Antarctica, over recent decades, Cryosphere, 7(1), 263–273, doi:10.5194/tc-7-263-2013, 2013.

Taylor, K. E., Stouffer, R. J., and Meehl, G. A.: An overview of CMIP5 and the experiment design, B. Am. Meteorol. Soc., 93, 485–498, doi:10.1175/BAMS-D-11-00094.1, 2012.

Yang, J. W., Han, Y., Orsi, A. J., Kim, S. J., Han, H., Ryu, Y., Jang, Y., Moon, J., Choi, T., Hur, S. Do and Ahn, J.: Surface Temperature in Twentieth Century at the Styx Glacier, Northern Victoria Land, Antarctica, From Borehole Thermometry, Geophys Res Lett, 45(18), 9834–9842, doi:10.1029/2018GL078770, 2018.

Zagorodnov, V., Nagornov, O., Scambos, T. A., Muto, A., Mosley-Thompson, E., Pettit, E. C. and Tyuflin, S.: Borehole temperatures reveal details of 20 th century warming at Bruce Plateau, Antarctic Peninsula, Cryosphere, 6(3), 675–686, doi:10.5194/tc-6-675-2012, 2012.

Klein, F. and Goosse, H.: Reconstructing East African rainfall and Indian Ocean sea surface temperatures over the last centuries using data assimilation, Clim. Dyn., 50, 3909–3929, https://doi.org/10.1007/s00382-017-3853-0, 2018.